# CBGBench: Fill in the Blank of Protein-Molecule Complex Binding Graph

**Haitao Lin**[1,3,†], **Guojiang Zhao**[2,†], **Odin Zhang**[3], **Yufei Huang**[1],
**Lirong Wu**[1], **Zicheng Liu**[1], **Cheng Tan**[1], **Zhifeng Gao**[2], **Stan Z. Li**[1,*]
[1]AI Lab, Research Center for Industries of the Future, Westlake University;
[2]DP Technology; [3]Zhejiang University; [†]Equal Contribution; [*]Corresponding Author;
`linhaitao@westlake.edu.cn`; `stan.zq.li@westlake.edu.cn`

## Abstract

Structure-based drug design (SBDD) aims to generate potential drugs that can bind to a target protein and is greatly expedited by the aid of AI techniques in generative models. However, a lack of systematic understanding persists due to the diverse settings, complex implementation, difficult reproducibility, and task singularity. Firstly, the absence of standardization can lead to unfair comparisons and inconclusive insights. To address this dilemma, we propose CBGBench, a comprehensive benchmark for SBDD, that unifies the task as a generative graph completion, analogous to fill-in-the-blank of the 3D complex binding graph. By categorizing existing methods based on their attributes, CBGBench facilitates a modular and extensible framework that implements cutting-edge methods. Secondly, a single de novo molecule generation task can hardly reflect their capabilities. To broaden the scope, we adapt these models to a range of tasks essential in drug design, considered sub-tasks within the graph fill-in-the-blank tasks. These tasks include the generative designation of de novo molecules, linkers, fragments, scaffolds, and sidechains, all conditioned on the structures of protein pockets. Our evaluations are conducted with fairness, encompassing comprehensive perspectives on interaction, chemical properties, geometry authenticity, and substructure validity. We further provide insights with analysis from empirical studies. Our results indicate that there is potential for further improvements on many tasks, with optimization in network architectures, and effective incorporation of chemical prior knowledge. Finally, to lower the barrier to entry and facilitate further developments in the field, we also provide a single codebase (`https://github.com/EDAPINENUT/CBGBench`) that unifies the discussed models, data pre-processing, training, sampling, and evaluation.

## 1 Introduction

The rapid and remarkable progress in Graph Neural Networks (GNNs) and generative models have advanced the bio-molecule design in these years (Jumper et al., 2021; Watson et al., 2023; Krishna et al., 2023). In structure-based drug design, AI-aided methods aim to learn the chemical space of the molecules that can bind to certain proteins as targets, which decreases the general chemical space of molecule compounds ($\sim 10^{60}$) to a more compact search space and enables the model to explore the potential binding drugs. Recent success in generative models such as diffusion models (Ho et al., 2020; Lipman et al., 2022; Song et al., 2020) further enhanced these methods to fully exploit the targeted chemical space, and AI-aided SBDD has been propelled into another prominence.

Despite the significance of the SBDD and the development of various approaches, there remains a lack of a comprehensive benchmark for this field covering various practical application scenarios. *On the one hand*, although different methods are proposed for the task, the experimental setup is not unified, and the evaluation protocol also differs. For example, GraphBP (Liu et al., 2022) use a different training and test split of Crossdocked2020 (Francoeur et al., 2020) from concurrent works like Pocket2Mol (Peng et al., 2022) and 3DSBDD (Luo et al., 2022); And DiffBP (Lin et al., 2022) and GraphBP employ `Gnina` (McNutt et al., 2021) instead of `AutoDock Vina` (Trott & Olson, 2009) as the evaluator for obtaining docking score as binding affinity. Besides, the complex implementation of models makes the modules coupled, so it is hard to figure out which proposed

module contributes to the improvements in performance. For example, though TARGETDIFF (Guan et al., 2023b) and DIFFBP are both diffusion-based methods, the first one uses EGNN (Satorras et al., 2022) to predict the ground-truth position of each atom in the molecule while the later one uses GVP (Jing et al., 2020) to remove the added noise in a score-based way. To have a systematic understanding of the designation for diffusion models in molecule generation, the network architectures should be fixed to keep the expressivity equal. *On the other hand*, the task of *de novo* generation of molecules is a branch of SBDD. There are also other important tasks, such as the design of molecular side chains in lead optimization, or linker design to combine functional fragments into a single, connected molecule (Igashov et al., 2022; Guan et al., 2023a). It is highly meaningful to explore whether these methods can be successfully transferred to applications in drug optimization.

In this way, here we propose **CBGBench** as a benchmark for SBDD, with the latest state-of-the-art methods included and integrated into a single codebase (Appendix. C). Firstly, we unify the generation of the binding molecule as a graph completion task, *i.e.* fill-in-the-blank of the 3D Complex Binding Graph. Therefore, the systematic categorization of existing methods is based on three dichotomies: (i) voxelized *v.s.* continuous position generation, (ii) one-shot *v.s.* auto-regressive generation, and (iii) domain-knowledge-based *v.s.* full-data-driven generation. As a result, the methods can be easily modularized and extensible in a unified framework. Secondly, CBGBench introduces a unified protocol with a comprehensive evaluation, including (i) chemical property, (ii) interaction, (iii) geometry and (iv) substructure analysis, with metrics extended and fair ranking considered. Moreover, thanks to our reformulation of the problem and extensible modular implementation, we can easily extend these methods to the other four tasks in lead optimization, including the designation of (i) linkers, (ii) fragments, (iii) side chains and (iv) scaffolds. Comprehensive evaluation is also conducted for these tasks, to explore the potential application value of the existing methods in lead optimization.

As a result, several brief conclusions can be reached as the following:

- CNN-based methods by modeling voxelized density maps as molecule representations remain highly competitive in target-aware molecule generation, and the diffusion-based ones achieve the state-of-the-art.

- For autoregressive methods, it is essential to enable the model to successfully capture the patterns of chemical bonds in training and generation of binding molecules.

- Prior knowledge has been incorporated into the model in recent works, but the improvements remain limited. Effective design of integrating physical and chemical domain knowledge remains a challenge, leaving substantial room for future research.

- Most evaluated methods can be well generalized to lead optimization. Empirical studies show that scaffold hopping is the most challenging task among them, while linker design is relatively the easiest. However, there is still a large space for improvements in these tasks.

- The conclusions drawn from the experimental results and with the evaluation protocol of CBGBench, are mainly consistent with those obtained on real-world disease targets, demonstrating generalizability and effectiveness of our benchmark.

## 2 BACKGROUND

### 2.1 PROBLEM STATEMENT

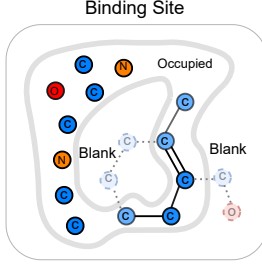

Figure 1: Filling the blank in the complex binding site.

For a binding system composed of a protein-molecule pair as $(\mathcal{P}, \mathcal{M})$, in which $\mathcal{P}$ contains $N_{\text{rec}}$ atoms of proteins and $\mathcal{M}$ contains $N_{\text{lig}}$ atoms of molecules, we represent the index set of the protein's atom as $\mathcal{I}_{\text{rec}}$ and the molecule's atoms as $\mathcal{I}_{\text{lig}}$, with $|\mathcal{I}_{\text{rec}}| = N_{\text{rec}}$ and $|\mathcal{I}_{\text{lig}}| = N_{\text{lig}}$. The binding graph can be regarded as a heterogenous graph. One subgraph is protein structures, as $\mathcal{P} = (\boldsymbol{V}_{\text{rec}}, \boldsymbol{E}_{\text{rec}})$, where $\boldsymbol{V}_{\text{rec}} = \{(a_i, \mathbf{x}_i, s_i)\}_{i \in \mathcal{I}_{\text{rec}}}$ is the node set, and $\boldsymbol{E}_{\text{rec}} = \{(i, j, e_{i,j})\}_{i,j \in \mathcal{I}_{\text{rec}}}$ is the edge set. Here, $a_i$ is the atom types with $a_i = 1, \dots, M$, $\mathbf{x}_i \in \mathbb{R}^3$ is the corresponding 3D position, and $s_i = 1, \dots, 20$ is the amino acid types that $i$-th atom belongs to; elements in edge set means there exists a bond between $i$ and $j$, with edge type $e_{i,j}$. The other graph is the molecule structures, written as $\mathcal{M} = (\boldsymbol{V}_{\text{lig}}, \boldsymbol{E}_{\text{lig}})$, where $\boldsymbol{V}_{\text{lig}} = \{(a_i, \mathbf{x}_i)\}_{i \in \mathcal{I}_{\text{lig}}}$ and $\boldsymbol{E}_{\text{lig}}$ is the edge set with the same form

Table 1: Categorization of included methods. LIGAN, 3DSBDD and VOXBIND model atom positions as discrete variables; One-shot methods maintain a constant atom number in a generation, like diffusion-based ones; FLAG and D3FG use fragment motifs, and DECOMPDIFF uses arm&scaffolding priors.

| Method | Continous Position | One-shot Generation | Domain Knowledge |
|---|---|---|---|
| LIGAN | ✗ | ✓ | ✗ |
| 3DSBDD | ✗ | ✗ | ✗ |
| POCKET2MOL | ✓ | ✗ | ✗ |
| GRAPHBP | ✓ | ✗ | ✗ |
| TARGETDIFF | ✓ | ✓ | ✗ |
| DIFFBP | ✓ | ✓ | ✗ |
| DIFFSBDD | ✓ | ✓ | ✗ |
| FLAG | ✓ | ✗ | ✓ |
| D3FG | ✓ | ✓ | ✓ |
| DECOMPDIFF | ✓ | ✓ | ✓ |
| MOLCRAFT | ✓ | ✓ | ✗ |
| VOXBIND | ✗ | ✓ | ✗ |

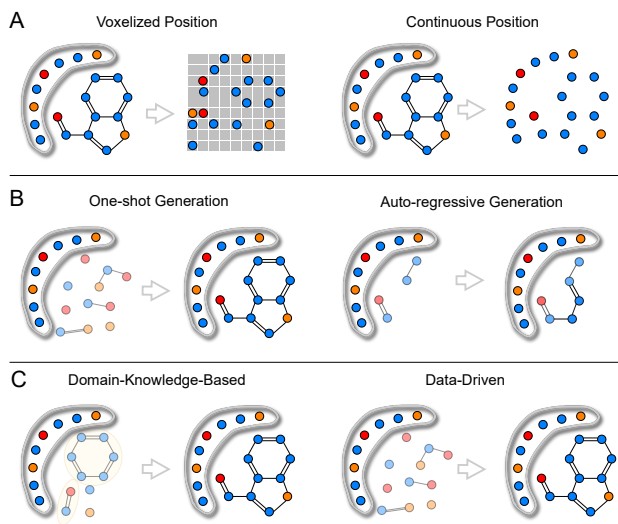

Figure 2: The demonstration of the classification criteria for the existing methods. The gray thick curves outline the contact surface of the protein, and the circles represent different types of atoms.

as $\boldsymbol{E}_{\mathrm{rec}}$. Besides, there are virtual edges that exist between protein and molecule, which make up cross-edge set $\boldsymbol{E}_{\mathrm{crs}}$. Denote the probabilistic model for binding graph by $p(\mathcal{M}, \mathcal{P})$. For the *de novo* molecule generation, the model aims to learn the probability of $p(\mathcal{M}|\mathcal{P})$, as to fill the blank of the protein pocket with atoms in a molecule, by using a generative model $p(\cdot)$.

A comparable analogy can be made that if we also divide the atoms in a molecule into known and to-be-generated parts, we can extend the aforementioned 'filling-in-the-blank' task as shown in Figure. 1. In specific, for the missing blanks, we write the node set for generation as $\mathcal{G} = (\boldsymbol{V}_{\mathrm{gen}}, \boldsymbol{E}_{\mathrm{gen}})$, and the known atoms in the molecule as elements in the context set $\mathcal{C} = (\boldsymbol{V}_{\mathrm{ctx}}, \boldsymbol{E}_{\mathrm{ctx}})$, in which $\mathcal{I}_{\mathrm{ctx}} = \mathcal{I}_{\mathrm{lig}} \setminus \mathcal{I}_{\mathrm{gen}}$. Therefore, *the generative model for filling the blanks of the missing parts of molecules aims to learn the probability of* $p(\mathcal{G}|\mathcal{C}, \mathcal{P})$. The newly-defined task of fill-in-the-partial-graph is of great significance in SBDD, especially in lead optimization, and we establish other four sub-tasks of it besides *de novo* design, with accessible datasets in Sec. 3.

## 2.2 RELATED WORK AND TAXONOMY

The SBDD methods are initially combined with deep neural networks on voxel grids, such as LIGAN (Masuda et al., 2020) generating atom voxelized density maps using VAE (Kingma & Welling, 2022) and incorporating Convolutional Neural Networks (CNNs) as the architecture, and 3DSBDD (Luo et al., 2022) predicts whether the position on grids is occupied with which type of atoms in auto-regressive ways with Graph Neural Networks (GNNs). Then, the development of Equivariant Graph Neural Networks (EGNNs) boosts the SBDD methods to directly generate the continuous 3D positions, such as POCKET2MOL (Peng et al., 2022) generating the molecules' atom types, positions, and the connected bonds and GRAPHBP (Liu et al., 2022) employing normalizing flows to generate these attributes, both in an auto-regressive way. The diffusion denoising probabilistic models (DDPM) (Ho et al., 2020) further propel the AI-aided SBDD methods, such as TARGETDIFF (Guan et al., 2023b), DIFFBP (Lin et al., 2022) and DIFFSBDD (Schneuing et al., 2022), inspired by EDM (Hoogeboom et al., 2022), generating full atoms' positions and element types, with different diffusion models. In recent years, domain knowledge has been used to constrain or guide the generation of binding drugs. For example, FLAG (Zhang et al., 2023b) and D3FG (Lin et al., 2023) use prior knowledge of fragment motifs to model the coarse structures, which generate fragments in molecules in one-shot and auto-regressive ways, respectively; And DECOMPDIFF (Guan et al., 2023c) harnesses virtual points searching for arm- and scaffold-clustering as prior knowledge, and using multivariate Gaussian process to model atom positions in different clusters, with validity guidance for sampling. More recently, thanks to technological breakthroughs brought by generative AI, more related approaches have been proposed, which not only build upon previous modeling

concepts but also incorporate new generative model techniques. MOLCRAFT (Qu et al., 2024), as the SBDD version of GEOBFN (Song et al., 2024), employs Bayesian Flow Network (Graves et al., 2024) as the variant of diffusion models, to address the continuous-discrete gap in modeling the elements' type and atoms' positions by applying continuous noise and smooth transformation; VOXBIND (Pinheiro et al., 2024a), following their pioneering exploration of VOXMOL (Pinheiro et al., 2024b), continues modeling the 3D voxelization of molecules in a diffusion denosing way, with walk-jump sampling method used to generate molecules. The latest FLEXSBDD (Zhang et al., 2024b) aims to generate the dynamic conformation of protein as well as design the binding molecules with Flow Mathcing techniques. Since the structure of protein targets may change significantly in FLEXSBDD, leading to different validation protocols, we will not conduct in-depth comparison here.

In this way, we can categorize these methods with the three standards:

- Whether the positions of atoms are generated in continuous 3D space or voxelized grids.
- Whether the generation process is auto-regressive or one-shot.
- Whether the domain knowledge is introduced to integrate extra prior into the model.

To better articulate, we classify them according to Table. 1, and Figure. 2 gives a simple demonstration of the criteria for taxonomy. A detailed review of these methods is given in Appendix. A.

## 3 TASK AND DATASET

For the *de novo* molecule generation, we follow the previous protocol to use Crossdocked2020 (Francoeur et al., 2020) and data preparation with splits proposed in LiGAN (Masuda et al., 2020) and 3DSBDD (Luo et al., 2022) as the training and test sets. Besides the *de novo* generation, the modularized methods can be extended to four subtasks which are branches of our defined fill-in-the-blank of the complex binding graph, including the target-aware linker, fragment, side chain, and scaffold design (Zhang et al., 2024a), leading to the generative targets $\mathcal{G}$ for the probabilistic model $p(\mathcal{G}|\mathcal{C}, \mathcal{P})$ to be the four components, and the molecular context $\mathcal{C}$ as the rest, as shown in Figure. 3. We demonstrate the significance of the four tasks in lead optimization and how the corresponding datasets are established below. The datasets are all established based on previous splits of Crossdocked2020 for a fair comparison.

**Linker Design** is a critical strategy in fragment-based drug discovery (Grenier et al., 2023). It focuses on creating linkers that connect two lead fragments and obtaining a complete molecule with enhanced affinity. Effective linker design can significantly influence the efficacy and pharmacokinetics of the drug, by ensuring that the molecule maintains the desired orientation and conformation when bound to its target (Erlanson et al., 2016). We define linkers following specific rules: (i) A linker should act as the intermediate structure connecting two fragments. (ii) A linker should contain at least two atoms on the shortest path between two fragments. (iii) Each connecting fragment must consist of more than five atoms.

**Fragment Growing** focus on expanding a fragment on the lead compound to better fill the binding pocket (Bancet et al., 2020). It also relates to adjusting pharmacological properties, such as enhancing solubility or reducing toxicity (Hung et al., 2009). We decompose the entire ligand into two fragments and select the smaller one for expansion based on specific criteria: (i) Each fragment must contain more than five atoms. (ii) The smaller fragment should be larger than half the size of the larger one.

**Side Chain Decoration** differs from fragment growing in that it permits modifications at multiple sites on the lead compound, whereas fragment growing typically modifies a single site. The taxonomy of arms-scaffold in previous work (Guan et al., 2023c) is similar to side-chain-scaffold, while we focus on a chemist-intuitive approach, Bemis-Murko decomposition (Bemis & Murcko, 1996), which treats all terminal non-cyclic structures as side chains.

**Scaffold Hopping** is introduced by Schneider et al. (1999), as a strategy in medicinal chemistry aiming to replace the core structure of molecules to explore more diverse chemical space or improve specific properties. While various empirical definitions exist (Sun et al., 2012), for consistency, the Bemis-Murko decomposition is utilized to define the scaffold structure.

Since chances are that there is no substructure in a molecule according to the discussed definition of the decomposition, we here give the instance number of training and test set for each task in Table. 2.

Table 2: The instance number of training and test split in the datasets for the four tasks and *De novo* generation. The decomposition is conducted separately in the training and test set of Cross-Docked2020 to avoid label leakage. Side chain decoration and scaffold hopping are two dual tasks, so the training and test set numbers are the same.

| Dataset | # Training | # Test |
|---|---|---|
| *De novo* | 99900 | 100 |
| Linker | 52685 | 43 |
| Fragment | 61379 | 61 |
| Side Chain | 70617 | 64 |
| Scaffold | 70617 | 64 |

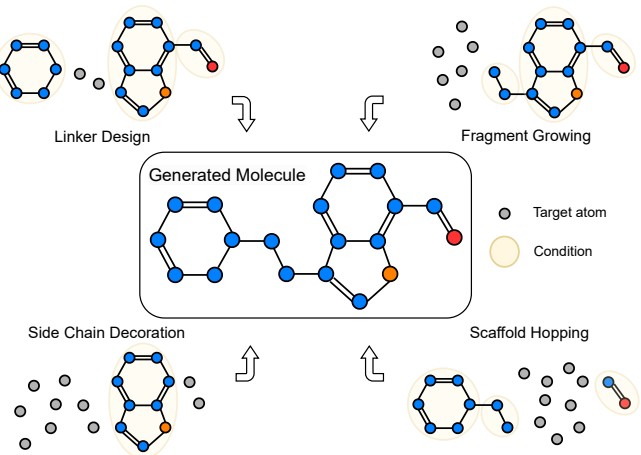

Figure 3: Demonstration of the four tasks.

## 4 EVALUATION PROTOCOL

In previous works, the evaluation usually focuses on two aspects including **interaction** and **chemical property**. While lots of works start to focus another two standards on molecule 3D structure's authenticity with **geometry** analysis and the 2D graphs reliability with **subsutructure** analysis, the evaluation protocol is usually not unified. In this way, we extend them in a unified protocol including the four aspects as the following:

**Substructure.** To evaluate whether the model succeeds in learning the 2D graph structures for drug molecules. We expand previous metrics to functional groups, as certain functional groups in the drugs act as pharmacophores, such as phenol and pyrimidine. We identify the 25 most frequently occurring functional groups by using EFG (Salmina et al., 2015) (Appendix. B.1 gives the details), and calculated mean frequency of each function group generated in a molecule and the overall multinomial distribution. In the same way, the two values can also be obtained according to rings and atom types. The generated frequency and distribution against the references lead to 6 metrics on substructure analysis including the Jensen-Shannon divergence as JSD and mean absolute error as MAE, listed as (i) $\mathbf{JSD_{at}}(\downarrow)$, (ii) $\mathbf{MAE_{at}}(\downarrow)$, (iii) $\mathbf{JSD_{rt}}(\downarrow)$, (iv) $\mathbf{MAE_{rt}}(\downarrow)$, (v) $\mathbf{JSD_{fg}}(\downarrow)$, and (vi) $\mathbf{MAE_{fg}}(\downarrow)$, where 'at' means atom type, 'rt' means ring type and 'fg' means functional group.

**Chemical Property.** We continue the evaluation of chemical properties from most previous works, including (i) $\mathbf{QED}(\uparrow)$ as quantitative estimation of drug-likeness; (ii) $\mathbf{SA}(\uparrow)$ as synthetic accessibility score; (iii) $\mathbf{LogP}$ represents the octanol-water partition coefficient, and in general LogP values should be between -0.4 and 5.6 to be good drug candidates (Ghose et al., 1999); (iv) $\mathbf{LPSK}(\uparrow)$ as the ratio of the generated drug molecules satisfying the Lipinski's rule of five.

**Interaction.** In evaluating the generated molecules' interaction with proteins, the binding affinity is usually the most important metric. Here we unify the affinity calculation with AutoDock Vina for fair comparison. However, previous studies show that larger molecule sizes will lead to a higher probability that the generated molecules can interact with the protein, resulting in higher predicted binding affinity (Abad-Zapatero & Metz, 2005; Hopkins et al., 2014). It is also an observation in this paper as the Pearson correlation of Vina Energy *v.s.* atom number reaches $-0.67$ (See Appendix. B.2). Besides, in some structures, there are more atoms in the protein participating in the interaction, which also causes the average to be unreasonable. Therefore, besides the commonly-used (i) $\boldsymbol{E_{\mathbf{vina}}}(\downarrow)$ as the mean Vina Energy, and (ii) $\mathbf{IMP}\%(\uparrow)$ as the improvements indicating the ratio of the generated molecule with lower binding energy than the reference, we use mean percent binding gap as (iii) $\mathbf{MPBG}(\uparrow)$ (Appendix. B.3 gives computation) and ligand binding efficacy as (iv) $\mathbf{LBE}(\uparrow)$, written as $\mathrm{LBE}_i = -\frac{E_{\mathrm{vina}}}{N_{\mathrm{lig}}}$, indicating how much does a single atom in the molecule contribute to the binding affinity, to eliminate the effects of molecule sizes. We give a detailed discussion on the reasonality of LBE in Appendix. B.4. Moreover, it is important for the generated molecules to keep the same or similar interaction patterns with proteins (Zhang et al., 2023a). Therefore, we aim to figure out whether the SBDD models can learn the microscopic interaction patterns hidden in the data of the 3D conformations. We use PLIP (Adasme et al., 2021; li Zuo et al., 2023; Salentin et al., 2015)

to characterize the 7 types of interactions and calculate per-pocket and overall JSD between the categorical distributions of generated types and reference types and MAE between the frequency of each generated types *v.s.* it of reference types, in which the frequency is defined as the average number of occurrences of each type of interaction produced by different single generated molecules binding to the same protein, leading to four metrics including (v) $\mathbf{JSD_{PP}}(\downarrow)$, (vi) $\mathbf{JSD_{OA}}(\downarrow)$, (vii) $\mathbf{MAE_{PP}}(\downarrow)$, (viii) $\mathbf{MAE_{OA}}(\downarrow)$, where 'PP' and 'OA' means per-pocket and overall respectively.

**Geometry.** The internal geometry is an important characteristic for distinguishing between general point clouds and molecular structures. The torsion angles (Jing et al., 2022; Swanson et al., 2023) are flexible geometries, while bond lengths and bond angles are static to reveal whether the generated molecules have realistic structures. Hence, we evaluate the overall JSD of bond length and angle distribution between reference and generated ones, written as (i) $\mathbf{JSD_{BL}}(\downarrow)$ and (ii) $\mathbf{JSD_{BA}}(\downarrow)$. The other perspective for validating structural resonability is the clash, the occurrence of which is defined when the van der Waals radii overlap by $\geq 0.4\text{Å}$ (Ramachandran et al., 2011). Hence, the ratio of number of atoms generating clashes with protein atoms to the total atom number is written as (iii) $\mathbf{Ratio_{cca}}(\downarrow)$ (cross clashes at atom level). Besides, the ratio of molecules with clashes as (iv) $\mathbf{Ratio_{cm}}(\downarrow)$ (molecule with clashes) are evaluated, which indicates if any atom in a molecule causes a clash, we consider that molecule to have a clash.

We here evaluate the 12 methods discussed in Table. 1, and use Friedman rank (Friedman, 1940; 1937; Wang et al., 2022) as the mean ranking method, to fairly compare the performance of different models in the four aspects. The ranking score is calculated by $(12 - \text{rank})$, and the final rank is according to the weighted mean ranking score. Appendix. B.3 gives detailed computation of the metrics, and their explanations as preliminary. The review of each method's evaluation aspect in the published papers is given in Apppendix. B.5.

## 5 BENCHMARK RESULT

### 5.1 *De novo* GENERATION

#### 5.1.1 SETUP

**Training.** We use the default configuration in each model's released codebase as the hyper-parameter, and set the training iteration number as 5,000,000 for fair comparison. It is noted that the loss of autoregressive methods exhibits a faster convergence rate of loss, typically requiring only a few tens of thousands of epochs to reach the final best checkpoint. To eliminate the effect brought about by the architecture of GNNs, in implementation, we use GVP (Jing et al., 2020) and EGNN (Satorras et al., 2022) with GAT (Veličković et al., 2018), as message-passing modules of auto-regressive and diffusion-based models, respectively. Especially, the GNN for encoding and decoding functional groups in D3FG is a combination of GAT and LoCS (Kofinas et al., 2022), CNN for processing atom density maps is four convolution blocks in LIGAN, and larger-scaled UNet for VOXBIND. The details in hyperparameter tuning and training strategy are given in Appendix. D.

**Evaluation.** Following the previous protocol, we generate 100 molecules per pocket in the test set, while not all the generated molecules are chemically valid (See Appendix. E.4), which will lead to <10,000 molecules. In affinity evaluation, we employ three modes of `AutoDock Vina`, including 'Score', 'Minimize', and 'Dock'. MPBG and LBE are added to the 'Vina Dock' mode. For Vina energy that is larger than 0, we think it is invalid, so we will not report it, and give the lowest ranking in the column of metrics. Moreover, LogP only provides a reference range for drug molecules, so we assigned a rank of 1 to the generated molecules within the range and 2 to those outside the range. In all the tables, values in **bold** are the best metric, and values with underline are the second and third.

#### 5.1.2 RESULT ANALYSIS

**Substructure.** Table. 3 gives results on 2D substructures, showing that (i) Overall, the auto-regressive models perform worse than the one-shot ones, especially in ring-type evaluation, where the former ones are more likely to generate triangular and tetrahedral rings. (ii) Diffusion-based models also exhibit advantages in atom type, according to the results of TARGETDIFF and MOLCRAFT, except for DIFFBP has difficulties in generating atoms with low occurrence frequencies. (iii) Besides VOXBIND, MOLCRAFT shows the overall superiority in complex functional group generation, as it states in addressing the mode collapse issues, while DECOMPDIFF and D3FG also exhibit advantages

Table 3: Results of substructure analysis.

| Metrics / Methods | Atom type | | Ring type | | Functional Group | | Rank |
|---|---|---|---|---|---|---|---|
| | JSD$_{at}$ | MAE$_{at}$ | JSD$_{rt}$ | MAE$_{rt}$ | JSD$_{fg}$ | MAE$_{fg}$ | |
| LiGAN | 0.1167 | 0.8680 | 0.3163 | 0.2701 | 0.2468 | 0.0378 | 6.50 |
| 3DSBDD | 0.0860 | 0.8444 | 0.3188 | 0.2457 | 0.2682 | 0.0494 | 6.50 |
| GraphBP | 0.1642 | 1.2266 | 0.5061 | 0.4382 | 0.6259 | 0.0705 | 11.33 |
| Pocket2Mol | 0.0916 | 1.0497 | 0.3550 | 0.3545 | 0.2961 | 0.0622 | 8.50 |
| TargetDiff | 0.0533 | **0.2399** | 0.2345 | 0.1559 | 0.2876 | 0.0441 | 4.00 |
| DiffSBDD | 0.0529 | 0.6316 | 0.3853 | 0.3437 | 0.5520 | 0.0710 | 8.33 |
| DiffBP | 0.2591 | 1.5491 | 0.4531 | 0.4068 | 0.5346 | 0.0670 | 10.83 |
| FLAG | 0.1032 | 1.7665 | 0.2432 | 0.3370 | 0.3634 | 0.0666 | 8.83 |
| D3FG | 0.0644 | 0.8154 | 0.1869 | 0.2204 | 0.2511 | 0.0516 | 5.17 |
| DecompDiff | **0.0431** | 0.3197 | 0.2431 | 0.2006 | **0.1916** | 0.0318 | 2.67 |
| MolCraft | 0.0490 | 0.3208 | 0.1196 | **0.0477** | 0.2469 | **0.0264** | 2.17 |
| VoxBind | 0.0942 | 0.3564 | **0.1053** | 0.0761 | 0.2401 | 0.0301 | 3.17 |

Table 4: Results of chemical property.

| | QED | LogP | SA | LPSK | Rank |
|---|---|---|---|---|---|
| LiGAN | 0.46 | **0.56** | 0.66 | 4.39 | 4.75 |
| 3DSBDD | 0.48 | **0.47** | 0.63 | 4.72 | 3.50 |
| GraphBP | 0.44 | **3.29** | 0.64 | 4.73 | 3.75 |
| Pocket2Mol | 0.39 | **2.39** | 0.65 | 4.58 | 4.75 |
| TargetDiff | 0.49 | **1.13** | 0.60 | 4.57 | 4.25 |
| DiffSBDD | 0.49 | **-0.15** | 0.34 | 4.89 | 3.50 |
| DiffBP | 0.47 | **5.27** | 0.59 | 4.47 | 5.25 |
| FLAG | 0.41 | **0.29** | 0.58 | **4.93** | 4.25 |
| D3FG | 0.49 | **1.56** | 0.66 | 4.84 | **2.00** |
| DecompDiff | 0.49 | **1.22** | 0.66 | 4.40 | 3.75 |
| MolCraft | 0.48 | **0.88** | **0.67** | 4.39 | 4.00 |
| VoxBind | **0.54** | **2.22** | 0.65 | 4.70 | 2.75 |

Table 5: Results of interacton analysis.

| Metrics / Methods | Vina Score | | Vina Min | | Vina Dock | | | | PLIP Interaction | | | | Rank |
|---|---|---|---|---|---|---|---|---|---|---|---|---|---|
| | $E_{vina}$ | IMP% | $E_{vina}$ | IMP% | $E_{vina}$ | IMP% | MPBG% | LBE | JSD$_{OA}$ | MAE$_{OA}$ | JSD$_{PP}$ | MAE$_{PP}$ | |
| LiGAN | **-6.47** | **62.13** | **-7.14** | **70.18** | -7.70 | **72.71** | 4.22 | 0.3897 | 0.0346 | 0.0905 | 0.1451 | **0.3416** | **2.91** |
| 3DSBDD | - | 3.99 | -3.75 | 17.98 | -6.45 | 31.46 | 9.18 | 0.3839 | 0.0392 | 0.0934 | 0.1733 | 0.4231 | 7.17 |
| GraphBP | - | 0.00 | - | 1.67 | -4.57 | 10.86 | -30.03 | 0.3200 | 0.0462 | 0.1625 | 0.2101 | 0.4835 | 11.33 |
| Pocket2Mol | -5.23 | 31.06 | -6.03 | 38.04 | -7.05 | 48.07 | -0.17 | **0.4115** | 0.0319 | 0.2455 | 0.1535 | 0.4152 | 5.67 |
| TargetDiff | -5.71 | 38.21 | -6.43 | 47.09 | -7.41 | 51.99 | 5.38 | 0.3537 | 0.0198 | 0.0600 | 0.1757 | 0.4687 | 4.67 |
| DiffSBDD | - | 12.67 | -2.15 | 22.24 | -5.53 | 29.76 | -23.51 | 0.2920 | 0.0333 | 0.1461 | 0.1777 | 0.5265 | 9.25 |
| DiffBP | - | 8.60 | - | 19.68 | -7.34 | 49.24 | 6.23 | 0.3481 | 0.0249 | 0.1430 | **0.1256** | 0.5639 | 7.41 |
| FLAG | - | 0.04 | - | 3.44 | -3.65 | 11.78 | -47.64 | 0.3319 | **0.0170** | 0.0277 | 0.2762 | 0.3976 | 9.00 |
| D3FG | - | 3.70 | -2.59 | 11.13 | -6.78 | 28.90 | -8.85 | 0.4009 | 0.0638 | **0.0135** | 0.1850 | 0.4641 | 8.17 |
| DecompDiff | -5.18 | 19.66 | -6.04 | 34.84 | -7.10 | 48.31 | -1.59 | 0.3460 | 0.0215 | 0.0769 | 0.1848 | 0.4369 | 6.08 |
| MolCraft | -6.15 | 54.25 | -6.99 | **56.43** | **-7.79** | 56.22 | 8.38 | 0.3638 | 0.0214 | 0.0780 | 0.1868 | 0.4574 | 3.75 |
| VoxBind | -6.16 | 41.80 | -6.82 | 50.02 | -7.68 | 52.91 | 9.89 | 0.3588 | 0.0257 | 0.0533 | 0.1850 | 0.4606 | 4.00 |

due to the incorporation of prior knowledge in ring types and functional groups. DiffBP and DiffSBDD perform poorly due to inconsistencies in generating complex fragments. For a detailed comparison, see Appendix. E.1.1

**Chemical Property.** The chemical property is calculated with 2D molecule graphs, so it can be greatly influenced by the molecule substructures. From Table. 4, we can conclude that (i) In terms of the four metrics, the differences among the compared methods are not significant. (ii) D3FG shows the best overall properties, with the competitive QED, SA, and LPSK.

**Interaction.** From Table. 5 shows that (i) LiGAN and VoxBind as CNN-based methods generate molecules that are initialized with high stability according to its competitive performance in Vina Score and Vina Min. It also performs well in Vina Dock mode, with a positive MPBG and high LBE, providing good consistency in interaction patterns. (ii) Auto-regressive methods except Pocket2Mol can hardly capture the pocket conditions and generate stably-binding molecules well, with very low IMP% in all Vina modes, while Pocket2Mol is the state-of-the-art auto-regressive method in *de novo* generation. (iii) In diffusion-based methods, MolCraft outperform the other competitors in overall interaction comparison, and TargetDiff and DecompDiff perform competitively since the difference between these two methods is minimal. DiffBP's performance in docking mode is comparable, but in other modes, it performs less satisfactorily. Performance of DiffSBDD is less than satisfactory. D3FG generates molecules with comparable Vina Energy but small atom numbers, leading to high LBE. For details, see Appendix. E.1.2.

**Geometry.** From Table. 6, conclusions can be drawn that (i) MolCraft generates overall the most realistic structures, according to the internal geometries, and DecompDiff is comparable. (ii) The molecule-protein clashes can usually be avoided by diffusion-based models. However, in the auto-regressive models, only Pocket2Mol avoids clashes. The high frequency of clashes in auto-regressive methods can be explained as follows: These models first identify frontier amino acids or atoms within the molecule. During the placing of new atoms, incorrect binding site localization leads to an erroneous direction for the molecule's auto-regressively growing path, causing it to extend inward towards the protein. For detailed results, see Appendix. E.1.3.

**Conclusion and Discussion.** From Table. 7 and the previous discussion, we conclude that

(i) The CNN-based methods like LiGAN and VoxBind are highly competitive, especially in aspects of **Interaction**, which explains why such methods in the field of drug design and molecule generation still prevail in recent years. This is partly attributed to *the fact*

Table 6: Results of geometry analysis.

| Metrics / Methods | Static Geometry | | Clash | | Rank |
|---|---|---|---|---|---|
| | $JSD_{BL}$ | $JSD_{BA}$ | $Ratio_{cca}$ | $Ratio_{cm}$ | |
| LiGAN | 0.4645 | 0.5673 | **0.0096** | **0.0718** | 5.75 |
| 3DSBDD | 0.5024 | 0.3904 | 0.2482 | 0.8683 | 8.75 |
| GraphBP | 0.5182 | 0.5645 | 0.8634 | 0.9974 | 11.50 |
| Pocket2Mol | 0.5433 | 0.4922 | 0.0576 | 0.4499 | 8.50 |
| TargetDiff | 0.2659 | 0.3769 | 0.0483 | 0.4920 | 4.50 |
| DiffSBDD | 0.3501 | 0.4588 | 0.1083 | 0.6578 | 7.25 |
| DiffBP | 0.3453 | 0.4621 | 0.0449 | 0.4077 | 5.25 |
| FLAG | 0.4215 | 0.4304 | 0.6777 | 0.9769 | 9.00 |
| D3FG | 0.3727 | 0.4700 | 0.2115 | 0.8571 | 8.50 |
| DecompDiff | 0.2576 | 0.3473 | 0.0462 | 0.5248 | 4.00 |
| MolCraft | **0.2250** | **0.2683** | 0.0264 | 0.2691 | **2.00** |
| VoxBind | 0.2701 | 0.3771 | 0.0103 | 0.1890 | 3.00 |

Table 7: Ranking scores and overall ranking.

| Weights / Methods | Substruc. 0.2 | Chem. 0.2 | Interact. 0.4 | Geom. 0.2 | Rank |
|---|---|---|---|---|---|
| LiGAN | 1.10 | 1.45 | 3.64 | 1.25 | 5 |
| 3DSBDD | 1.10 | 1.70 | 1.93 | 0.65 | 7 |
| GraphBP | 0.13 | 1.65 | 0.27 | 0.10 | 11 |
| Pocket2Mol | 0.70 | 1.45 | 2.53 | 0.70 | 7 |
| TargetDiff | 1.60 | 1.55 | 2.93 | 1.50 | 3 |
| DiffSBDD | 0.73 | 1.70 | 1.10 | 0.95 | 9 |
| DiffBP | 0.23 | 1.35 | 1.84 | 1.35 | 8 |
| FLAG | 0.63 | 1.55 | 1.20 | 0.60 | 10 |
| D3FG | 1.37 | 2.00 | 1.53 | 0.70 | 6 |
| DecompDiff | 1.87 | 1.65 | 2.57 | 1.60 | 4 |
| MolCraft | 1.97 | 1.60 | 3.30 | 2.00 | **1** |
| VoxBind | 1.77 | 1.85 | 3.20 | 1.80 | 2 |

*that CNNs have an advantage over GNNs in perceiving many-body patterns within a single filter* (Townshend et al., 2021). As a result, it encourages further research into GNNs for 3D point clouds to develop architectures that can match the expressivity of CNNs.

(ii) In GNN-based methods, MOLCRAFT achieves the best overall performance, with TARGETD-IFF closely following. In contrast, DECOMPDIFF and D3FG as the variants of TARGETDIFF that incorporate domain knowledge show some degeneration in performance. It reveals that the current incorporation of physicochemical priors can hardly improve the quality of generated molecules. *Effectively integrating domain knowledge to guide the model to generate structurally sound molecules remains a challenge.* For example, atom clashes are very common in generated molecules. While DECOMPDIFF employs the prior guidance for this, the problem has still not been fully solved. Integrating domain knowledge into the training process may mitigate this issue (Huang et al., 2024a; Adams & Coley, 2023).

(iii) Only POCKET2MOL as an auto-regressive method achieves competitive results, which we attribute to the following reasons: First, it utilizes chemical bonds to constrain atoms to grow orderly along chemical bonds rather than to grow based on distance to the pocket as in DIFFBP. Second, it simultaneously predicts the bond types and employs contrastive learning by sampling positive and negative instances of atom positions as real and fake bonds, which is not fully considered by FLAG, enhancing the model's ability to perceive chemical bond patterns. Therefore, we believe that *enabling the autoregressive methods model to successfully capture the patterns of chemical bonds is very essential*.

## 5.2 EXTENSION ON SUBTASKS

**Setup.** Lead optimization is to strengthen the function or property of the binding molecules by remodeling the existing partial 3D graph of molecules. We show the interaction analysis and chemical property in the main text and omit the interaction pattern analysis since maintaining the patterns is not necessary for lead optimization. The domain-knowledge-based methods can hardly be transferred for these tasks since different tasks require different priors, so we have not compared them here. Besides, the voxelize-based methods are not easily extended to these tasks, and we regard the transferring of methods like LiGAN and 3DSBDD as future work. Hence, we here compare 6 methods that model atoms' continuous positions with GNNs and have not employed domain knowledge. When training, since the number of training instances is smaller, we use the pretrained models on *de novo* generation and finetune the auto-regressive models with 1,000,000 iterations. For diffusion-based models with one-shot generation, we train them from scratch because the zero-center-of-mass (Satorras et al., 2021) technique is shifted from employing protein geometric centers to using molecule context's ones. For detailed results, please refer to Appendix E.2.

**Conclusion and Discussion.** From Table. 8 and the previous discussion, we conclude that

(i) Overall, the performance gap among these methods in the lead optimization is not as pronounced compared to *de novo* generation. MOLCRAFT, TARGET and POCKET2MOL maintain the good performance. Notably, in the evaluation of the stability of initial poses, the other three also complete the binding graph near the given partial molecular conformations well, according to the columns of Vina Score and Vina Min. The applicability of GRAPHBP also reflects the *Argument. (ii)* in our *de novo* geometry analysis: The failure of GraphBP mainly stems from the difficulty in locating the correct atoms for auto-regressive growth.

Table 8: Results of subtasks for lead optimization.

| Tasks | Methods | Vina Score | | Vina Min | | Vina Dock | | | | Chem. Prop. | | | | Rank |
|---|---|---|---|---|---|---|---|---|---|---|---|---|---|---|
| | | $E_{\text{vina}}$ | IMP% | $E_{\text{vina}}$ | IMP% | $E_{\text{vina}}$ | IMP% | MPBG% | LBE | QED | LogP | SA | LPSK | |
| Linker | GraphBP | - | 5.63 | -0.97 | 12.43 | -7.51 | 28.18 | -7.36 | 0.3288 | 0.41 | **0.86** | **0.70** | 3.60 | 4.75 |
| | Pocket2Mol | -6.89 | 18.99 | -7.19 | 25.04 | -8.07 | 37.22 | -3.85 | 0.3276 | **0.45** | **1.93** | 0.67 | **4.25** | 2.83 |
| | TargetDiff | **-7.22** | 36.11 | -7.60 | 41.31 | -8.49 | 50.73 | 2.61 | 0.2993 | 0.39 | **1.63** | 0.61 | 3.17 | 3.17 |
| | DiffSBDD | -5.64 | 11.06 | -6.38 | 19.15 | -7.88 | 34.97 | -4.42 | 0.3110 | 0.42 | **1.14** | 0.66 | 4.10 | 4.25 |
| | DiffBP | -6.27 | 35.49 | -7.19 | 36.80 | -8.74 | 54.33 | 6.60 | 0.3078 | 0.43 | 3.45 | 0.55 | 4.01 | 3.25 |
| | MolCraft | -7.13 | 38.80 | **-7.81** | 42.56 | -8.81 | 58.44 | 5.09 | **0.3334** | 0.43 | 1.06 | 0.67 | 4.14 | **1.50** |
| Fragment | GraphBP | -5.54 | 4.86 | -6.28 | 9.78 | -7.16 | 16.21 | -11.88 | **0.3749** | 0.54 | 0.87 | 0.66 | 4.66 | 3.58 |
| | Pocket2Mol | **-6.87** | 22.78 | **-7.61** | 37.45 | **-8.33** | 54.05 | -0.28 | 0.3310 | 0.46 | 1.02 | 0.63 | 4.07 | **2.00** |
| | TargetDiff | -6.06 | 24.56 | -6.78 | 30.43 | -7.96 | 42.00 | -2.38 | 0.3003 | 0.45 | 1.43 | 0.58 | 4.28 | 3.33 |
| | DiffSBDD | -4.64 | 19.14 | -5.84 | 28.90 | -7.66 | 37.18 | -6.67 | 0.3076 | 0.47 | 0.73 | 0.58 | 4.39 | 4.17 |
| | DiffBP | -4.51 | 22.31 | -6.18 | 29.52 | -7.90 | 45.70 | -1.92 | 0.2952 | 0.46 | 2.24 | 0.49 | 4.30 | 4.08 |
| | MolCraft | -6.75 | 21.12 | -7.06 | 36.07 | -7.92 | 43.02 | **-0.09** | 0.3236 | 0.46 | 1.27 | 0.51 | 4.64 | 2.58 |
| Side chain | GraphBP | 5.01 | 10.15 | -5.43 | 11.46 | -6.14 | 9.71 | -11.05 | **0.4459** | 0.61 | 1.93 | 0.76 | 4.93 | 3.75 |
| | Pocket2Mol | -5.99 | 22.26 | -6.56 | 33.29 | -7.26 | 41.04 | -4.34 | 0.3600 | 0.49 | 0.21 | 0.65 | 4.20 | 2.91 |
| | TargetDiff | -5.80 | 23.90 | -6.50 | 35.81 | -7.40 | 46.87 | -2.55 | 0.3213 | 0.48 | 0.88 | 0.60 | 4.41 | 2.58 |
| | DiffSBDD | -4.43 | 15.12 | -5.99 | 30.23 | -7.58 | 44.09 | -9.38 | 0.3178 | 0.43 | 1.20 | 0.65 | 4.03 | 3.83 |
| | DiffBP | -4.61 | 14.31 | -5.73 | 24.29 | -7.03 | 38.96 | -7.38 | 0.3143 | 0.49 | 1.29 | 0.56 | 4.50 | 4.25 |
| | MolCraft | **-6.10** | 24.10 | **-6.64** | 35.58 | -7.49 | 41.67 | -3.12 | 0.3227 | 0.44 | 1.22 | 0.61 | 4.36 | 2.42 |
| Scaffold | GraphBP | - | 0.00 | - | 0.06 | -3.90 | 0.99 | -50.62 | **0.3797** | 0.43 | 0.14 | 0.76 | 4.98 | 4.16 |
| | Pocket2Mol | -4.80 | 16.84 | -5.71 | 23.08 | -6.89 | 38.18 | -8.07 | 0.3378 | 0.43 | 0.91 | 0.64 | 4.48 | 2.42 |
| | TargetDiff | **-5.52** | 31.47 | **-5.86** | 34.39 | -7.06 | 44.32 | -6.22 | 0.3038 | 0.43 | 0.89 | 0.59 | 4.26 | **2.00** |
| | DiffSBDD | -3.85 | 18.44 | -4.90 | 22.12 | -6.81 | 34.98 | -10.23 | 0.2985 | 0.42 | -0.13 | 0.53 | 4.29 | 4.17 |
| | DiffBP | -2.09 | 13.89 | -4.35 | 16.84 | -6.46 | 32.43 | -12.14 | 0.3025 | **0.43** | 3.37 | 0.56 | 4.44 | 4.17 |
| | MolCraft | -4.71 | 14.03 | -5.35 | 30.16 | -7.02 | 43.53 | -7.34 | 0.3146 | 0.42 | 1.01 | 0.56 | 4.55 | 2.83 |

(ii) In these tasks, scaffold hopping is the most challenging task, as the improvement of generated molecules relative to the reference is minimal; Linker design is relatively the easiest. Notably, in scaffolding, there are still some methods that fail.

(iii) There is a large space for improvements in these tasks, since the MPBG% metrics are usually negative, indicating that in most cases, the molecule is not optimized. Some edge-cutting techniques such as DPO (Rafailov et al., 2024; Cheng et al., 2024; Gu et al., 2024) and ITA (Yang et al., 2020; Kong et al., 2023; Lin et al., 2024) may be used to augment the optimized molecules as supervision signals for the models.

Additionally, several points warrant further detailed design. For instance, in linker design, the fragments of the molecule that need to be connected may change orientation due to variations in the linker (Guan et al., 2023a); In side chain decoration, the protein's side chains, which are the main components in interaction, should be generated together with the molecule's side chains to achieve a stable conformation as the entire complex (Luo et al., 2023; Huang et al., 2024b).

## 5.3 CASE STUDY ON REAL-WORLD DISEASE TARGETS

**Introduction.** In order to verify whether the included methods can generalize to pharmaceutic targets related to disease and the applicability of CBGBench to real-world scenarios, we use the pretrained model in *de novo* generation, and apply them to two proteins belonging to the G-Protein-Coupled Receptor (GPCR) family: ARDB1 (beta-1 adrenergic receptor) and DRD3 (dopamine receptor D3). ARDB1 participates in the regulation of various physiological processes by responding to the neurotrans mitter epinephrine (adrenaline) and norepinephrine, and drugs that selectively activate or block this receptor are used in the treatment of various cardiovascular conditions. For DRD3, it is primarily expressed in the brain, particularly in areas such as the limbic system and the ventral striatum with functions of mediating the effects of the neurotrans mitter dopamine in the central nervous system.

**Setup.** On these targets, there are molecules reported active to them experimentally, we here randomly select 200 of them for each target and conduct two kinds of experiments. Firstly, we try to figure out if the model can generate binding molecules that have similar chemical distributions with the actives. We use extended connectivity fingerprint (ECFP) (Rogers & Hahn, 2010) to get the molecule fingerprint and employ t-SNE (van der Maaten & Hinton, 2008) for 2-dimensional visualization. Secondly, we aim to find out the distribution of Vina Docking Energy and LBE of the generated and the actives as the metrics for binding affinities. Eight methods except for voxelized-based ones (inflexible to extend) and DECOMPDIFF (requiring complex data-preprocessing for domain knowledge) are tested on the two targets. Besides, we select 100 molecules randomly from GEOM-DRUG (Axelrod & Gómez-Bombarelli, 2022), as a randomized control sample set.

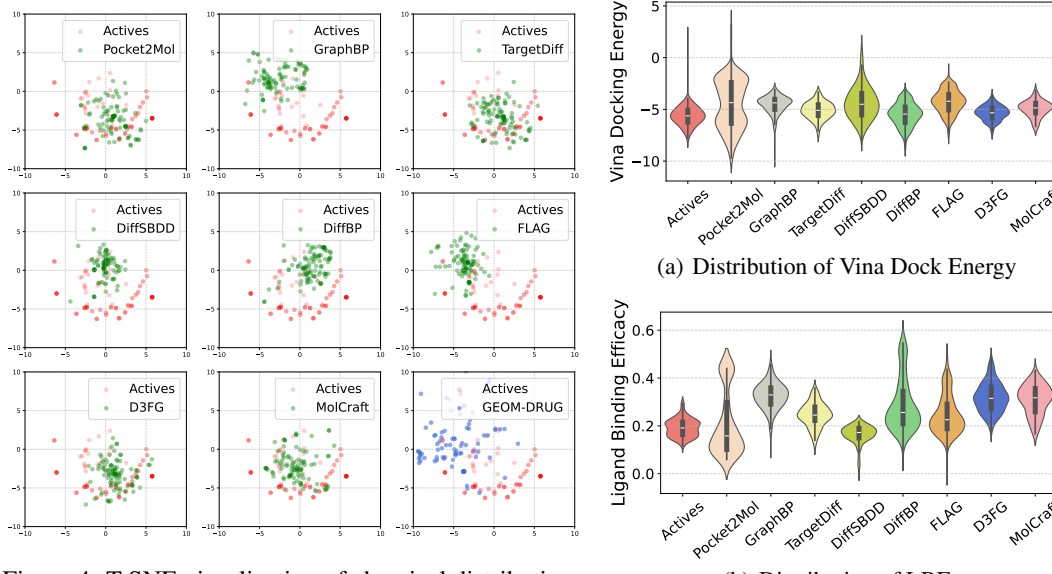

Figure 4: T-SNE visualization of chemical distributions of generated and active molecules on ADRB1.

(a) Distribution of Vina Dock Energy

(b) Distribution of LBE

Figure 5: Distribution of binding affinities.

**Conculusion and Discussion.** Figure. 4 and Figure. 5 gives the results on target ADRB1.

(i) In Figure. 4, we can see that POCKET2MOL, TARGETDIFF, D3FG, and MOLCRAFT have better consistency in the chemical distribution of molecules, as evidenced by a greater degree of overlap with the actives. In addition, in comparison to randomly selected molecules in GEOM-DRUG, these models show different preferences in generating binding molecules since the clustering center in the chemical space differs.

(ii) Figure. 5 shows that in generating molecules based on the real-world target ADRB1 related to hypertension and heart failure, D3FG exhibits superior performance. TARGETDIFF and MOLCRAFT perform comparably.

These conclusions are essentially consistent with the conclusion in Sec. 5.1, reflecting that the established evaluation protocols exhibit **consistency and generalizability on real-world disease target data**, especially in evaluating the binding affinity through Vina Dock. Besides, it is worth noting that DIFFBP, GRAPHBP, and POCKET2MOL can possibly generate molecules with small atom numbers and high LBE. This indicates that they have the potential to excel in lead discovery on ADRB1, as a good lead should possess good synthesizability and modifiability and smaller molecular weight. For DRD3, please refer to Appendix. E.3 for details.

## 6 CONCLUSION AND LIMITATION

CBGBench unifies the tasks of SBDD and lead optimization into a fill-in-the-blank 3D binding graph, comprehensively categorizes and modularizes existing methods, and integrates them into a unified codebase for fair comparison. Additionally, it extends existing evaluation protocols by incorporating more aspects with metrics, addressing the issue of incomplete and diverse evaluation process. Extensive experiments including case studies on disease targets for molecule generation give insightful conclusions and identify future research directions.

However, there are certain limitations. Firstly, this codebase is based on GNNs, so the voxelized-grid-based methods with CNNs have not been included. Engineering the integration of these types of methods will be a focus of our future work. Secondly, due to the inability to use wet lab experiments for validation, most metrics are obtained through computational methods, some of which are considered unable to accurately reflect the chemical properties of molecules, such as SA as a very important metrics but its deficiency is usually ignored by molecule design methods (Luo et al., 2024). Therefore, how to utilize AI to assist in accurate metric calculation will also be a key focus of our future research. Besides, several works (Zheng et al., 2024) emphasized the potential of 1D/2D molecule designation in these year, which is also an important aspect of research interest.

ACKNOWLEDGEMENTS

This work was supported by National Science and Technology Major Project (No. 2022ZD0115101), National Natural Science Foundation of China Project (No. 624B2115, No. U21A20427), Project (No. WU2022A009) from the Center of Synthetic Biology and Integrated Bioengineering of Westlake University, Project (No. WU2023C019) from the Westlake University Industries of the Future Research Funding.

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

## A REVIEW ON THE METHODS' TIMELINE

Here we give a brief review of the 12 included methods in chronological order, with Table. **??** giving some standardized information on these methods.

- LIGAN is firstly proposed to incorporate a generative model for the SBDD task, in which the encoder and the decoder are 3-layer CNNs, and the Variational Auto-Encoder is employed to model the probability of the atom's density map in a molecule. It split the training and test set in `CrossDocked2020`.

- Due to the rise of Graph Neural Networks (GNNs), 3DSBDD subsequently adopted GNNs for modeling this task. At that time, GNNs were more adept at handling pairwise information, so an autoregressive graph network modeling approach was chosen as an effective method for molecular modeling. However, the development of graph networks for 3D tasks remained slow, leading 3DSBDD to continue modeling molecules using voxelized maps. 3DSBDD leverages graph networks to predict the voxel position that the next atom will occupy, as well as the atom type, thereby generating molecules in an autoregressive manner. In addition, 3DSBDD further standardized the evaluation protocol of the experiment by using Vina Docking Energy to assess the stability of the generated molecules.

- The rise of EGNN has enabled autoregressive modeling of continuous atomic coordinates in the SBDD task while preserving equivariance and invariance properties. Following this development, GRAPHBP adopted the EGNN architecture, using a depth-first search approach to rank the molecules from near to far on the protein surface. This allows for autoregressive molecular generation starting from the protein frontier and ultimately modeling a complete molecule. However, GRAPHBP did not consider critical chemical priors such as bond lengths and bond angles between atoms, making it challenging to capture the internal patterns of molecules. The evaluation is based on Gnina Docking, different from 3DSBDD.

- Meanwhile, POCKET2MOL directly modeled atomic coordinates using GVP. Compared to GRAPHBP, POCKET2MOL took molecular bonds into account and employed contrastive learning, where false atoms were used to regularize the coordinates and bonding of real atoms. This significantly improved the stability of the generated molecules. The evaluation follows 3DSBDD.

- The rise of generative models, particularly the widespread application of diffusion models in the field of images, has drawn attention to these cutting-edge generative techniques in the molecular domain as well. Diffusion models are inspired by particle systems in statistical physics, which aligns closely with the task of generating atomic coordinates in molecules. DIFFSBDD, as pioneering works in applying diffusion models to the SBDD task, have also achieved significant results. DIFFSBDD followed most of the concepts from EDM, such as the zero-center-of-mass technique, and was tested and generalized on `Binding MOAD` beyond `CrossDocked2020`.

- Meanwhile, DIFFBP also proposed to use the Gaussian Diffusion model on atoms' positions and masked type diffusion for elements' types. DIFFBP analyzed the drawbacks of autoregressive models from the perspective of the Boltzmann energy distribution in statistical physics and observed that molecular weight severely affects docking performance, being the first to adopt a grouping approach for model comparison.

- After that, following previous explorations like GEODIFF and EDM in molecular (conformation) generation, TARGETDIFF innovatively introduced a diffusion model-based SBDD method, achieving new state-of-the-art (SOTA) performance. For validation, TARGETDIFF first proposed using Vina Score and Vina Min mode to assess the stability of the generated molecules' initial states. Additionally, the embeddings learned by the model can also be used for affinity prediction.

- On the other hand, fragment-based generative methods are also in the early stages of exploration. FLAG follows the autoregressive generation approach of POCKET2MOL, breaking molecules into fragments and individual atoms, allowing for the generation of complex molecules with fewer steps. However, its modeling approach lacks sufficient consideration of chemical bond participation, which is a key reason for its suboptimal molecular generation performance.

- Afterward, D3FG combined fragment-based generation with the diffusion method, leveraging protein generation techniques to establish a diffusion model that predicts the orientation matrix, central coordinates, and fragment types based on fragment information. The model then uses linkers to connect the molecule fragments. Additionally, D3FG enables molecular optimization by replacing fragments that are either prominent or insufficiently contributing.

- In contrast to D3FG, which treats fragments as rigid bodies, DECOMPDIFF considers fragments as flexible structures, thereby decomposing molecules into arms and scaffolds, and performing multivariate Gaussian diffusion within different decomposition clusters. Additionally, DECOMPDIFF employs non-intersection guidance to mitigate the issue of atomic overlap between the molecule and protein amino acids.

- In addition, work on voxelized maps is steadily progressing. VOXBIND builds on the pioneering work of VOXMOL, using diffusion-based denoising modeling on the voxelized density map of molecules. During the generation steps, it incorporates wj-sampling to reduce computational complexity, while also producing structurally stable molecules.

- More recently, MOLCRAFT was proposed, utilizing the Bayes Flow Network, which is designed for modeling probabilistic densities' parameters. This allows molecules to avoid the difficulty of coupling the two modalities: the discrete variable of atomic types and the continuous positions of atoms in the molecule. It effectively addresses issues such as mode collapse and has demonstrated outstanding performance as the most recent SOTA model.

Recently, several cocurrent benchmark works are also proposed for SBDD tasks, such as Zheng et al. (2024), which aims to demonstrate that 1D/2D molecular generation methods remain highly competitive compared to 3D approaches, which is insightful for researchers to rethink that if 3D-molecule design is required for the tasks. In contrast, our work focuses solely on generative 3D SBDD tasks. Moreover, we provide a comprehensive evaluation of 3D molecular generation methods, incorporating over 10 3D-based methods compared to the five included in Zheng et al. (2024).

Table 9: A brief review of the included methods and task adaptation in our CBGBench.

| Method | Time | Generative Model | Network Architecture | Prior Knowledge | Evaluation Datasets | Task adaptation (CBGBench) |
|---|---|---|---|---|---|---|
| LIGAN | Oct. 2020 | VAE | CNN | None | CrossDocked2020 | De novo design |
| 3DSBDD | Mar. 2022 | Auto-regressive | GNN | None | CrossDocked2020 | De novo design / Lead optimization |
| GRAPHBP | Apr. 2022 | Auto-regressive | EGNN | None | CrossDocked2020 | De novo Design / Lead optimization |
| POCKET2MOL | May. 2022 | Auto-regressive | GVP | None | CrossDocked2020 | De novo Design / Lead optimization |
| DIFFSBDD | Oct. 2022 | DDPM | EGNN | None | CrossDocked2020 / Binding MOAD | De novo Design / Lead optimization |
| DIFFBP | Nov. 2022 | DDPM | GVP | None | CrossDocked2020 | De novo design / Lead optimization |
| TARGETDIFF | Feb. 2023 | DDPM | EGNN | None | CrossDocked2020 | De novo design / Lead optimization |
| FLAG | Feb. 2023 | Auto-regressive | GVP | Fragment | CrossDocked2020 | De novo design |
| D3FG | May. 2023 | DDPM | EGNN + IPA | Fragment | CrossDocked2020 | De novo design |
| DECOMPDIFF | Feb. 2024 | DDPM | EGNN | Arm&Scaffold | CrossDocked2020 | De novo design |
| MOLCRAFT | Apr. 2024 | BFN | EGNN | None | CrossDocked2020 | De novo design / Lead optimization |
| VOXBIND | May. 2024 | DDPM | CNN | None | CrossDocked2020 | De novo design |

# B  SUPPLEMENTARY EVALUATION DETAILS

## B.1  INCLUDED FUNCTIONAL GROUPS

Here we show the functional groups included in this paper, in which we follow D3FG and give a demonstration of them in Table. 10.

Table 10: The included functional groups in CBGBench. 'T' is the occurrence times of the functional group in the datasets (100,000 ligands). 'A,B,C' are the framing node index.

| Smiles | 2D graph | 3D structures | A | B | C | T |
|--------|----------|---------------|---|---|---|---|
| c1ccccc1 | | | 1 | 0 | 2 | 131148 |
| NC=O | | | 1 | 0 | 2 | 49023 |
| O=CO | | | 1 | 0 | 2 | 39863 |
| c1ccncc1 | | | 3 | 2 | 4 | 15115 |
| c1ncc2nc[nH]c2n1 | | | 7 | 3 | 6 | 11369 |
| NS(=O)=O | | | 1 | 0 | 2 | 10121 |
| O=P(O)(O)O | | | 1 | 0 | 2 | 7451 |
| OCO | | | 1 | 0 | 2 | 6405 |
| c1cncnc1 | | | 3 | 2 | 4 | 5965 |
| c1cn[nH]c1 | | | 2 | 3 | 1 | 5404 |

| Smiles | 2D graph | 3D structures | A | B | C | T |
|---|---|---|---|---|---|---|
| O=P(O)O | | | 0 | 1 | center(2,3) | 5271 |
| c1ccc2ccccc2c1 | | | 3 | 2 | 4 | 4742 |
| c1ccsc1 | | | 3 | 2 | 4 | 4334 |
| N=CN | | | 1 | 0 | 2 | 4315 |
| NC(N)=O | | | 2 | 1 | 3 | 4167 |
| O=c1cc[nH]c(=O)[nH]1 | | | 7 | 1 | 5 | 4145 |
| c1ccc2ncccc2c1 | | | 3 | 2 | 4 | 3519 |
| c1cscn1 | | | 2 | 3 | 1 | 3466 |
| c1ccc2[nH]cnc2c1 | | | 5 | 4 | 6 | 3462 |
| c1c[nH]cn1 | | | 3 | 2 | 4 | 2964 |
| O=[N+][O-] | | | 1 | 0 | 2 | 2702 |
| O=CNO | | | 1 | 0 | 2 | 2477 |
| NC(=O)O | | | 1 | 0 | 2 | 2438 |
| O=S=O | | | 1 | 0 | 2 | 2375 |
| c1ccc2[nH]ccc2c1 | | | 3 | 4 | 2 | 2301 |

## B.2 LBE ANALYSIS

The LBE is motivated by the following phenomena in Fig. B.2, which shows that the Vina Energy has an strong correlation with the atom number. Besides, in Appendix B4, we give a more detailed analysis of LBE.

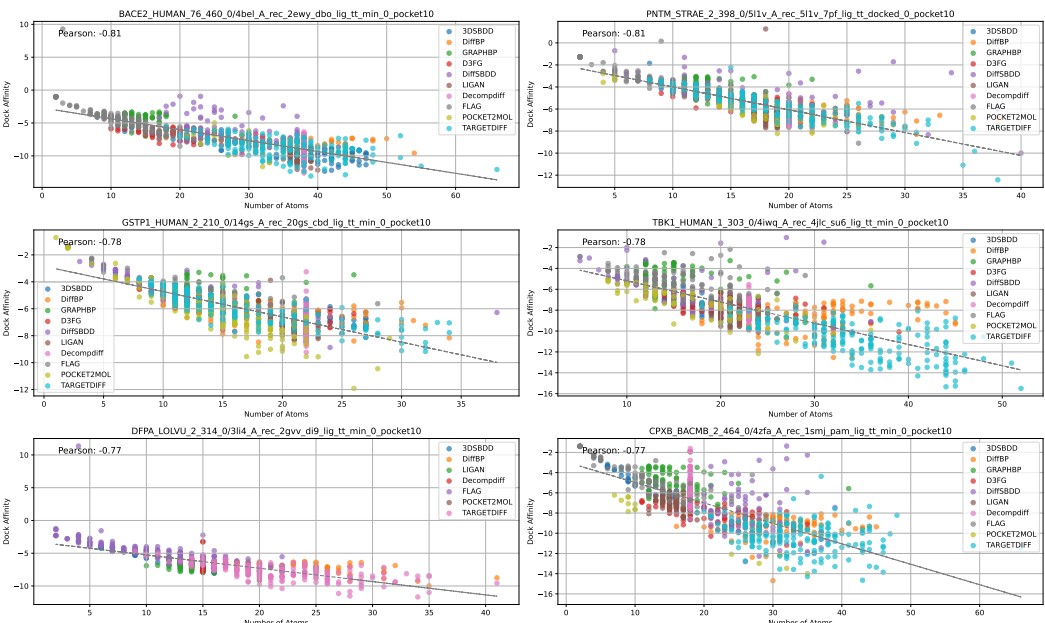

Figure 6: A demonstration of how the molecule size affects the binding energy. The 6 pockets that are most linear-correlated with atom number.

## B.3 METRIC CALCULATION

The metric of MPBG can be written as $\mathrm{MPBG}_j = \mathtt{Mean}_i(\frac{E_{i,\mathrm{gen}} - E_{\mathrm{ref}}}{E_{\mathrm{ref}}} \times 100\%)$, in which $i$ is the indicator of the generated molecules in a single protein and $\mathtt{Mean}_i(\cdot)$ calculates average along indicator $i$, with $\mathbf{MPBG} = \mathtt{Mean}_j(\mathrm{MPBG}_j)$. The MPBG is a per-pocket metric that is calculated within a single pocket, which differs from other averaging metrics.

For chemical property calculation, we use functions in RDKit (RDKit, online), and note that the SA is the normalized one employed in the previous studies.

For interaction pattern analysis, we employ PLIP, which considers 7 kinds of interaction types including 'hydrophobic interactions', 'hydrogen bonds', 'water bridges', '$\pi$-stacks', '$\pi$-cation interactions', 'halogen bonds' and 'metal complexes'.

For MAE metrics that are related to the molecule sizes, the generative model tends to achieve higher scores when the atom number is close to that of the reference. We suppose it is reasonable, considering that the size of the molecule itself is determined by factors such as the size and shape of the pocket, and these characteristics should be captured by the models.

## B.4 DISCUSSION ON LBE.

*- Is it possible that smaller molecules exhibit higher efficacy, thus leading LBE to be meaningless? (Reynolds et al., 2008)*

**Meaning.** The introduction of Ligand Binding Efficiency (LBE) is motivated by observations in Appendix B.2, where we identified a strong positive correlation between the size of the generated molecules and the Vina docking energy. To address this size-related bias, LBE was incorporated as an additional metric. This approach is consistent with the basic principles of energy calculation,

Table 11: Ratio of generated molecule's size and the corresponding LBE.

| Method | Mol Size | [1,3] | (3,10] | (10,20] | (20,30] | >30 |
|---|---|---|---|---|---|---|
| LiGAN | Ratio(%) | 0.01 | 13.61 | 40.09 | 36.77 | 9.59 |
|  | LBE | 0.37 | 0.42 | 0.44 | 0.35 | 0.30 |
| 3DSBDD | Ratio(%) | 0.01 | 15.48 | 54.44 | 22.63 | 7.42 |
|  | LBE | 0.39 | 0.42 | 0.41 | 0.33 | 0.31 |
| GRAPHBP | Ratio(%) | 0.00 | 0.77 | 91.29 | 6.07 | 1.84 |
|  | LBE | - | 0.36 | 0.33 | 0.21 | 0.20 |
| POCKET2MOL | Ratio(%) | 0.08 | 24.22 | 48.83 | 20.19 | 6.64 |
|  | LBE | 0.43 | 0.45 | 0.44 | 0.33 | 0.30 |
| TARGETDIFF | Ratio(%) | 0.00 | 10.24 | 28.17 | 39.51 | 22.05 |
|  | LBE | - | 0.44 | 0.44 | 0.31 | 0.29 |
| DIFFSBDD | Ratio(%) | 0.13 | 21.54 | 41.43 | 28.01 | 8.86 |
|  | LBE | 0.37 | 0.38 | 0.30 | 0.24 | 0.22 |
| DIFFBP | Ratio(%) | 0.00 | 7.29 | 32.19 | 43.08 | 17.42 |
|  | LBE | - | 0.41 | 0.42 | 0.31 | 0.29 |
| FLAG | Ratio(%) | 0.04 | 39.01 | 54.29 | 6.52 | 0.13 |
|  | LBE | 0.40 | 0.35 | 0.33 | 0.24 | 0.19 |
| D3FG | Ratio(%) | 0.00 | 8.31 | 61.70 | 27.72 | 2.55 |
|  | LBE | - | 0.44 | 0.45 | 0.32 | 0.25 |
| DECOMPDIFF | Ratio(%) | 0.00 | 9.99 | 35.50 | 33.98 | 20.51 |
|  | LBE | - | 0.38 | 0.38 | 0.33 | 0.30 |
| MOLCRAFT | Ratio(%) | 0.00 | 9.18 | 30.97 | 39.62 | 20.21 |
|  | LBE | - | 0.44 | 0.41 | 0.33 | 0.29 |
| VOXBIND | Ratio(%) | 0.00 | 6.10 | 36.64 | 43.39 | 15.85 |
|  | LBE | - | 0.43 | 0.41 | 0.34 | 0.26 |

where the total binding energy is derived from the sum of interaction energies contributed by each atom, i.e. $E_{\text{vina}} \sim E_{\text{bind}} = \sum_{i=1}^{N_{\text{atom}}} E_i$. Thus, using LBE as a measure of the average per-atom contribution to binding affinity is both logical and appropriate, as $\text{LBE} = -\frac{E_{\text{bind}}}{N_{\text{atom}}}$. Furthermore, numerous studies in medicinal chemistry have highlighted Ligand Efficiency as a critical evaluation criterion, advocating for its widespread use as an effective metric for assessing molecular binding affinity (Hopkins et al., 2014; Kenny, 2017; Murray et al., 2010).

**Resonability.** It can be usually concluded from Reynolds et al. (2008) that 'smaller molecules tend to have higher LBE', while we believe this conclusion is incorrect. We claim that different protein pockets exhibit a preference for molecule sizes. If this conclusion were true, it would imply that molecules consisting of a single atom would have the optimal LBE? To explore this issue, in Table. 11, we give statistics of molecule sizes v.s. LBE of methods evaluated. It shows that such molecules with extreme sizes (less than 4) that make the LBE meaningless are scarcely generated since they have a small ratio. Ligands with sizes less than 20 usually have better LBE because of the preference of the pockets, as a consequence of protein binding sites being limited in their size, thus generating the preference of binding molecules' sizes. Larger ligand with sizes of more than 20 gets decreased LBE because it becomes increasingly difficult to form optimal interactions with every site on the protein without introducing unfavorable ligand strain.

**Little Effects of Extreme Samples.** Besides, we use a weighted ranking with equal weights as previous benchmarks (Wang et al., 2022), to give an unbiased evaluation. Given that LBE accounts for one-twelfth of the total weight in interaction analysis, the impact of failed samples in LBE on calculating the LBE metrics and final ranking is minimal. Therefore, we claim that the inclusion of a minimal number of invalid LBE calculations will not render the overall ranking ineffective.

### B.5 PREVIOUSLY-USED METRICS FOR EVALUATION

The aspects of evaluation are commonly used with different methods, while they hardly give a very comprehensive evaluation. We give a brief review of each method's evaluation on these aspects in Table. 12.

Table 12: Different methods' evaluation aspects in the published papers.

| Method | Substructure | Geometry | Chem Property | Interaction |
|---|---|---|---|---|
| LiGAN | Figure. S6 | Figure. S7, S8, S9 | Figure. 3, S3, S4, S5 | Figure. S13, S14 |
| Pocket2Mol | Table. 2 | Table. 3; Figure. 4 | Table. 1 | Table. 1 |
| GraphBP | - | Table. 2; Figure. 5 | - | Table. 1; Figure. 2 |
| TargetDiff | Table. 2 | Table. 1; Figure. 2 | Table. 3 | Table. 3, Figure. 4 |
| DiffBP | Table. 3 | - | Table. 2 | Table. 1 |
| DiffSBDD | - | Figure. 8, 9 | Table. 1 | Table. 1, 2; Figure. 2 |
| FLAG | Table. 3 | Table. 2; Figure. 4 | Table. 1 | Table. 1 |
| D3FG | Table. 1, 3; Figure. 3 | Table. 2 | Table. 4 | Table. 4 |
| DecompDiff | - | Table. 1, 2; Figure. 3 | Table. 3 | Table. 3 |
| MolCraft | Table. 1 | Table. 2 | Table. 2 | Table. 2 |
| VoxBind | - | Figure. 7 | Table. 1 | Table. 1; Figure. 5 |

## C CODEBASE STRUCTURE

In this section, we provide an overview of the codebase structure of CBGBench, where four abstract layers are adopted. The layers include the core layer, algorithm layer, chemistry layer, and API layer in the bottom-up direction as shown in Fig. 7. This codebase is licensed under the Apache License, Version 2.0. It provides a robust and flexible framework for building and evaluating graph neural network models for structure-based drug design and lead optimization.

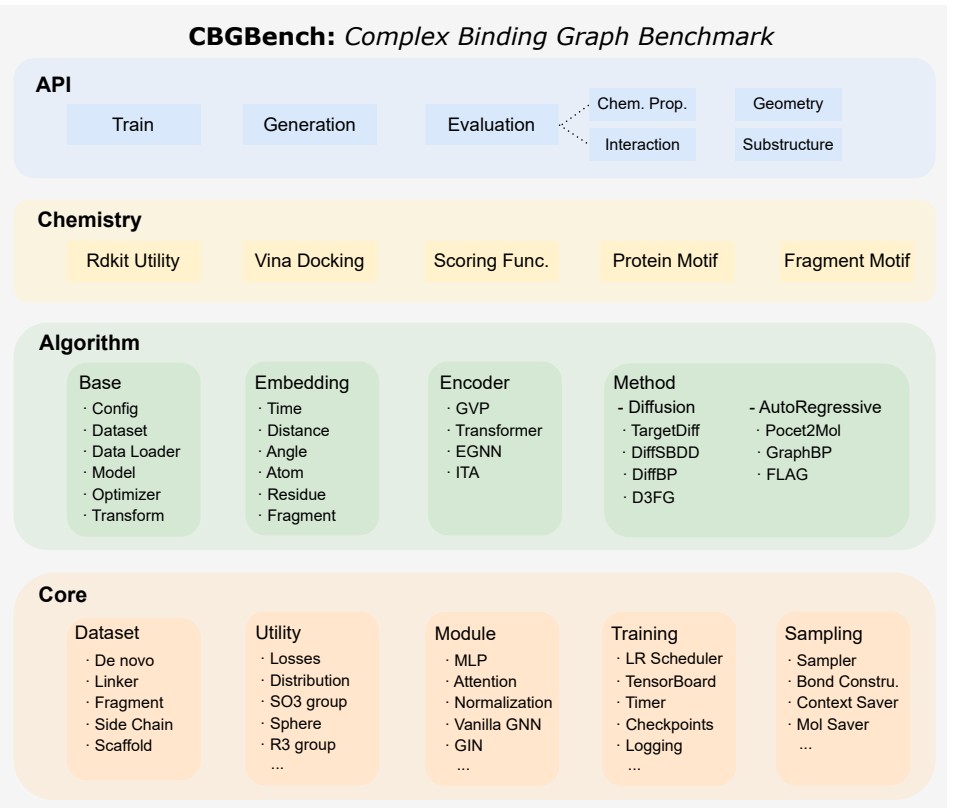

Figure 7: Structure of CBGBench Codebase, consisting of 4 layers. The core layer provides the common functions, datasets, and modules for CBG methods. The algorithm layer mainly implements the prevailing CBG algorithms. The Chemistry layer is used for data preprocessing and evaluation.

**Core Layer.** In the core layer, we implement the commonly used core functions for training and sampling CBG algorithms. Besides, the code regarding datasets, data loaders, and basic modules used in CBGBench is also provided in the core layer.

**Algorithm Layer.**    In the algorithm layer, we first implement the base class for CBG algorithms, where we initialize the datasets, data loaders, and basic modules from the core layer. We modularize each method, first using embedding layers to project the input into high-dimensional space, and then employing different 3D equivariant GNNs to construct the algorithm, including Diffusion-based and Auto-regressive-based ones. We further abstract the algorithms, enabling better code reuse and making it easier to implement new algorithms. The voxel-based methods are not included because our framework is mostly based on EGNNs, which will be added to the codebase as future work, and DecompDiff requires different data preprocessing. Based on this, we support 7 core CBG algorithms. More algorithms are expected to be added through continued extension.

**Chemistry Layer.**   The chemistry layer is mostly built with post-processing utilities and evaluators such as atom type, bond angle, and docking modules. Besides, several prior decomposition functions are used as molecule parsers, and the molecule and protein fragment motifs serve as prior knowledge.

**API Layer.**    We wrap the core functions and algorithms in CBGBench in the API layer as a public Python package. It is friendly for users from different backgrounds who want to employ CGB algorithms in new applications. Training and sampling can be done in only a few lines of code. In addition, we provide the configuration files of all algorithms supported in CBGBench with detailed parameter settings, allowing the reproduction of the results.

# D    EXPERIMENTAL IMPLEMENTATION DETAILS

Experiments are conducted based on Pytorch 2.0.1 on a hardware platform with Intel(R) Xeon(R) Gold 6240R @ 2.40GHz CPU and NVIDIA A100 GPU. We give the detailed parameters for training the included models in Table. 13. Note that most of the hyperparameters' combinations are directly the officially-provided ones. Specifically, The methods included in our codebase are either self-trained or trained by us when the official repository does not provide a pretrained checkpoint. For methods that are not included, we validate them using the pretrained models that have been provided. In the Lead-Optimization tasks, the finetuning parameters are the same.

Table 13: The hyper-parameters in for training and sampling molecules for the SBDD methods. Different from GNN-based methods, CNN-based methods usually have a different hidden dimension size, so we report the highest ones. 'Num steps' are the sampling steps which is usually used in diffusion-based methods, including DDPMs and BFNs. For other hyper-parameters in detail, please read the config files provided in the supplementary materials from `./configs/denovo/train`.

| Method | Batch Size | Hidden Dim | Layer num | Learning rate | Optimizer | Num Steps |
|---|---|---|---|---|---|---|
| LIGAN | 8 | 128 | 4 | 1E-05 | RMSprop | / |
| 3DSBDD | 4 | 256 | 6 | 1E-04 | Adam | / |
| GRAPHBP | 16 | 64 | 6 | 1E-04 | Adam | / |
| POCKET2MOL | 8 | 256 | 6 | 1E-05 | Adam | / |
| TARGETDIFF | 4 | 128 | 9 | 5E-04 | Adam | 1000 |
| DIFFSBDD | 16 | 256 | 6 | 1E-03 | Adam | 500 |
| DIFFBP | 16 | 256 | 6 | 1E-04 | Adam | 1000 |
| FLAG | 4 | 256 | 6 | 1E-04 | Adam | / |
| D3FG | 16 | 128 | 6 | 1E-04 | Adam | 1000 |
| DECOMPDIFF | 4 | 128 | 6 | 5E-04 | Adam | 1000 |
| MOLCRAFT | 8 | 128 | 9 | 5E-04 | Adam | 1000 |
| VOXBIND | 32 | 512 | 4 | 1E-05 | AdamW | 100 |

# E    EXPERIMENTAL RESULT DETAILS

## E.1    DE NOVO GENERATION

### E.1.1    SUBSTRUCTURE

Table. 14 gives the generated molecules' atom distribution in detail. It shows DIFFBP and GRAPHBP tend to generate more C atoms and have a low probability of generating uncommon atoms. In contrast, D3FG and FLAG, which directly use a motif library, have a higher probability of generating these

uncommon atoms. Table. 15 gives the generated molecules' ring distribution in detail. DIFFBP and DIFFSBDD perform poorly due to significant inconsistencies in generating large complex fragments. For example, they also generate a large number of unreasonable triangular and tetrahedral rings. Similarly, these methods also fall short in generating complex functional groups, as shown in Table. 16. This leads to their subpar performance in substructure generation.

Table 14: Distribution of different atom types across different methods.

| Method | C | N | O | F | P | S | Cl |
|---|---|---|---|---|---|---|---|
| REF. | 0.6715 | 0.1170 | 0.1696 | 0.0131 | 0.0111 | 0.0112 | 0.0064 |
| LIGAN | 0.6477 | 0.0775 | 0.2492 | 0.0005 | 0.0224 | 0.0019 | 0.0008 |
| 3DSBDD | 0.6941 | 0.1311 | 0.1651 | 0.0025 | 0.0063 | 0.0010 | 0.0000 |
| GRAPHBP | 0.8610 | 0.0397 | 0.0868 | 0.0036 | 0.0040 | 0.0039 | 0.0010 |
| POCKET2MOL | 0.7623 | 0.0855 | 0.1413 | 0.0025 | 0.0044 | 0.0027 | 0.0013 |
| TARGETDIFF | 0.6935 | 0.0896 | 0.1924 | 0.0110 | 0.0059 | 0.0052 | 0.0025 |
| DIFFSBDD | 0.7000 | 0.1154 | 0.1611 | 0.0081 | 0.0017 | 0.0093 | 0.0031 |
| DIFFBP | 0.9178 | 0.0030 | 0.0792 | 0.0000 | 0.0000 | 0.0000 | 0.0000 |
| FLAG | 0.5585 | 0.1341 | 0.2077 | 0.0265 | 0.0312 | 0.0347 | 0.0074 |
| D3FG | 0.7336 | 0.1158 | 0.1286 | 0.0056 | 0.0035 | 0.0088 | 0.0040 |
| DECOMPDIFF | 0.6762 | 0.0978 | 0.1927 | 0.0064 | 0.0149 | 0.0088 | 0.0033 |
| MOLCRAFT | 0.6735 | 0.0917 | 0.2056 | 0.0103 | 0.0094 | 0.0058 | 0.0035 |
| VOXBIND | 0.7359 | 0.1083 | 0.1390 | 0.0000 | 0.0046 | 0.0120 | 0.7382 |

Table 15: Distribution of different ring sizes across various methods.

| Method | 3 | 4 | 5 | 6 | 7 | 8 |
|---|---|---|---|---|---|---|
| REF. | 0.0130 | 0.0020 | 0.2855 | 0.6894 | 0.0098 | 0.0003 |
| LIGAN | 0.2238 | 0.0698 | 0.2599 | 0.4049 | 0.0171 | 0.0096 |
| 3DSBDD | 0.2970 | 0.0007 | 0.1538 | 0.5114 | 0.0181 | 0.0116 |
| GRAPHBP | 0.0000 | 0.2429 | 0.1922 | 0.1765 | 0.1533 | 0.1113 |
| POCKET2MOL | 0.0000 | 0.1585 | 0.1822 | 0.4373 | 0.1410 | 0.0478 |
| TARGETDIFF | 0.0000 | 0.0188 | 0.2856 | 0.4918 | 0.1209 | 0.0298 |
| DIFFSBDD | 0.2842 | 0.0330 | 0.2818 | 0.2854 | 0.0718 | 0.0193 |
| DIFFBP | 0.0000 | 0.2195 | 0.2371 | 0.2215 | 0.1417 | 0.0707 |
| FLAG | 0.0000 | 0.0682 | 0.2716 | 0.5228 | 0.0996 | 0.0231 |
| D3FG | 0.0000 | 0.0201 | 0.2477 | 0.5966 | 0.0756 | 0.0283 |
| DECOMPDIFF | 0.0302 | 0.0378 | 0.3407 | 0.4386 | 0.1137 | 0.0196 |
| MOLCRAFT | 0.0000 | 0.0022 | 0.2494 | 0.6822 | 0.0489 | 0.0072 |
| VOXBIND | 0.0000 | 0.0062 | 0.2042 | 0.7566 | 0.0232 | 0.0021 |

Table 16: Distribution of the top ten functional groups across different methods.

| Method | c1ccccc1 | NC=O | O=CO | c1ccncc1 | c1ncc2nc[nH]c2n1 | NS(=O)=O | O=P(O)(O)O | OCO | c1cncnc1 | c1cn[nH]c1 |
|---|---|---|---|---|---|---|---|---|---|---|
| REF. | 0.3920 | 0.1465 | 0.1192 | 0.0452 | 0.0340 | 0.0303 | 0.0223 | 0.0191 | 0.0178 | 0.0162 |
| LIGAN | 0.3464 | 0.0998 | 0.1549 | 0.0546 | 0.1028 | 0.0127 | 0.0490 | 0.0897 | 0.0197 | 0.0000 |
| 3DSBDD | 0.3109 | 0.1488 | 0.1219 | 0.0769 | 0.0090 | 0.0000 | 0.0814 | 0.0915 | 0.0148 | 0.0291 |
| GRAPHBP | 0.0133 | 0.1330 | 0.1888 | 0.0000 | 0.0000 | 0.0000 | 0.0000 | 0.6064 | 0.0000 | 0.0000 |
| POCKET2MOL | 0.3794 | 0.1098 | 0.2906 | 0.0305 | 0.0000 | 0.0000 | 0.0150 | 0.0964 | 0.0030 | 0.0000 |
| TARGETDIFF | 0.2729 | 0.1520 | 0.3085 | 0.0427 | 0.0001 | 0.0000 | 0.0241 | 0.0855 | 0.0069 | 0.0065 |
| DIFFSBDD | 0.0073 | 0.1985 | 0.5787 | 0.0194 | 0.0000 | 0.0073 | 0.0000 | 0.0145 | 0.0000 | 0.0000 |
| DIFFBP | 0.1962 | 0.0108 | 0.4440 | 0.0010 | 0.0000 | 0.0000 | 0.0000 | 0.3289 | 0.0000 | 0.0000 |
| FLAG | 0.2223 | 0.1880 | 0.1736 | 0.0365 | 0.0009 | 0.0000 | 0.0014 | 0.2471 | 0.0135 | 0.0014 |
| D3FG | 0.3002 | 0.1597 | 0.2078 | 0.0323 | 0.0114 | 0.0000 | 0.0212 | 0.0751 | 0.0410 | 0.0000 |
| DECOMPDIFF | 0.3173 | 0.1660 | 0.2125 | 0.0527 | 0.0094 | 0.0057 | 0.0418 | 0.0443 | 0.0134 | 0.0128 |
| MOLCRAFT | 0.3960 | 0.1798 | 0.2361 | 0.0377 | 0.0001 | 0.0000 | 0.0362 | 0.0344 | 0.0053 | 0.0002 |
| VOXBIND | 0.4705 | 0.1138 | 0.1663 | 0.0762 | 0.0001 | 0.0000 | 0.0191 | 0.0334 | 0.0110 | 0.0000 |

### E.1.2 INTERACTION

**PLIP Interaction Pattern.** Table. 17 and 18 provide detailed interaction pattern analysis. Most models captured good interaction patterns, generating a substantial amount of hydrophobic and hydrogen interactions. DiffSBDD excessively generated hydrophobic interactions. Additionally, for interactions such as $\pi$-stacking, $\pi$-cation, and halogen interactions, which do not exist in the reference molecules, the models could probabilistically generate molecules capable of producing such interactions with the protein.

### E.1.3 GEOMETRY

Here we give detailed bond length and bond angle distribution of the evaluated method's generated molecules and the reference, in Table. 19 and 20. It shows that the

Table 17: Frequency of interaction type.

| Method | hydrophobic | hydrogen | water bridge | $\pi$-stacks | $\pi$-cation | halogen | metal |
|---|---|---|---|---|---|---|---|
| REF. | 3.0000 | 3.0000 | 0.0000 | 0.0000 | 0.0000 | 0.0000 | 0.0000 |
| LIGAN | 3.1354 | 3.5078 | 0.0000 | 0.1045 | 0.0441 | 0.0034 | 0.0000 |
| 3DSBDD | 3.1237 | 3.0853 | 0.0000 | 0.1405 | 0.0434 | 0.0053 | 0.0000 |
| GRAPHBP | 5.2008 | 0.8039 | 0.0000 | 0.0007 | 0.0046 | 0.0060 | 0.0000 |
| POCKET2MOL | 4.9366 | 1.8840 | 0.0000 | 0.0176 | 0.0081 | 0.0045 | 0.0000 |
| TARGETDIFF | 4.2444 | 3.5309 | 0.0000 | 0.1159 | 0.0377 | 0.0384 | 0.0000 |
| DIFFSBDD | 3.9009 | 3.3202 | 0.0000 | 0.0986 | 0.0338 | 0.0279 | 0.0000 |
| DIFFBP | 4.1027 | 1.4830 | 0.0000 | 0.0213 | 0.0039 | 0.0000 | 0.0000 |
| FLAG | 1.9319 | 2.1490 | 0.0000 | 0.0134 | 0.0185 | 0.0528 | 0.0000 |
| D3FG | 4.3149 | 2.4718 | 0.0000 | 0.0607 | 0.0260 | 0.0200 | 0.0000 |
| DECOMPDIFF | 3.9284 | 3.4531 | 0.0000 | 0.1177 | 0.0436 | 0.0290 | 0.0000 |
| MOLCRAFT | 4.0147 | 3.9456 | 0.0000 | 0.2038 | 0.0667 | 0.0377 | 0.0000 |
| VOXBIND | 4.4920 | 2.8395 | 0.0000 | 0.1659 | 0.0599 | 0.0001 | 0.0000 |

Table 18: Distibution of interaction type.

| Method | hydrophobic | hydrogen | water bridge | $\pi$-stacks | $\pi$-cation | halogen | metal |
|---|---|---|---|---|---|---|---|
| REF. | 0.5000 | 0.5000 | 0.0000 | 0.0000 | 0.0000 | 0.0000 | 0.0000 |
| LIGAN | 0.4614 | 0.5162 | 0.0000 | 0.0154 | 0.0065 | 0.0005 | 0.0000 |
| 3DSBDD | 0.4882 | 0.4822 | 0.0000 | 0.0220 | 0.0068 | 0.0008 | 0.0000 |
| GRAPHBP | 0.8645 | 0.1336 | 0.0000 | 0.0001 | 0.0008 | 0.0010 | 0.0000 |
| POCKET2MOL | 0.7206 | 0.2750 | 0.0000 | 0.0026 | 0.0012 | 0.0007 | 0.0000 |
| TARGETDIFF | 0.5327 | 0.4432 | 0.0000 | 0.0146 | 0.0047 | 0.0048 | 0.0000 |
| DIFFSBDD | 0.5285 | 0.4498 | 0.0000 | 0.0134 | 0.0046 | 0.0038 | 0.0000 |
| DIFFBP | 0.6579 | 0.3398 | 0.0000 | 0.0020 | 0.0004 | 0.0000 | 0.0000 |
| FLAG | 0.4638 | 0.5159 | 0.0000 | 0.0032 | 0.0045 | 0.0127 | 0.0000 |
| D3FG | 0.6259 | 0.3586 | 0.0000 | 0.0088 | 0.0038 | 0.0029 | 0.0000 |
| DECOMPDIFF | 0.5188 | 0.4560 | 0.0000 | 0.0155 | 0.0058 | 0.0038 | 0.0000 |
| MOLCRAFT | 0.4855 | 0.4771 | 0.0000 | 0.0246 | 0.0080 | 0.0045 | 0.0000 |
| VOXBIND | 0.5943 | 0.3757 | 0.0000 | 0.0219 | 0.0007 | 0.2012 | 0.0000 |

Table 19: JSD Bond Length Comparisons across different methods.

| METHOD | C-C | C-N | C-O | C=C | C=N | C=O |
|---|---|---|---|---|---|---|
| LIGAN | 0.4986 | 0.4146 | 0.4560 | 0.4807 | 0.4776 | 0.4595 |
| 3DSBDD | 0.2090 | 0.4258 | 0.5478 | 0.5170 | 0.6701 | 0.6448 |
| GRAPHBP | 0.5038 | 0.4231 | 0.4973 | 0.6235 | 0.4629 | 0.5986 |
| POCKET2MOL | 0.5667 | 0.5698 | 0.5433 | 0.4787 | 0.5989 | 0.5025 |
| TARGETDIFF | 0.3101 | 0.2490 | 0.3072 | 0.1715 | 0.1944 | 0.3629 |
| DIFFSBDD | 0.3841 | 0.3708 | 0.3291 | 0.3043 | 0.3473 | 0.3647 |
| DIFFBP | 0.5704 | 0.5256 | 0.5090 | 0.6161 | 0.6314 | 0.5296 |
| FLAG | 0.3460 | 0.3770 | 0.4433 | 0.4872 | 0.4464 | 0.4292 |
| D3GF | 0.4244 | 0.3227 | 0.3895 | 0.3860 | 0.3570 | 0.3566 |
| DECOMPDIFF | 0.2562 | 0.2007 | 0.2361 | 0.2590 | 0.2844 | 0.3091 |
| MOLCRAFT | 0.2473 | 0.1732 | 0.2341 | 0.3040 | 0.1459 | 0.2250 |
| VOXBIND | 0.3335 | 0.2577 | 0.3507 | 0.1991 | 0.1459 | 0.3334 |

Table 20: JSD Bond Angle Comparisons across different methods.

| $JSD_{BA}$ | LIGAN | 3DSBDD | GRAPHBP | POCKET2MOL | TARGETDIFF | DIFFSBDD | DIFFBP | FLAG | D3GF | DECOMPDIFF | MOLCRAFT | VOXBIND |
|---|---|---|---|---|---|---|---|---|---|---|---|---|
| C#C-C | 0.6704 | 0.4838 | 0.7507 | 0.6477 | 0.6845 | 0.6788 | 0.7204 | 0.4591 | 0.7027 | 0.8174 | 0.4922 | 0.6275 |
| C-C#N | 0.8151 | 0.2980 | 0.8326 | 0.5830 | 0.7437 | 0.7388 | 0.7928 | 0.3461 | 0.7120 | 0.7254 | 0.2252 | 0.7320 |
| C-C-C | 0.5260 | 0.2189 | 0.5015 | 0.4663 | 0.2955 | 0.3825 | 0.5234 | 0.3439 | 0.3703 | 0.2306 | 0.1926 | 0.2742 |
| C-C-N | 0.5102 | 0.2934 | 0.4975 | 0.4790 | 0.2738 | 0.4265 | 0.5189 | 0.3650 | 0.3592 | 0.1987 | 0.1097 | 0.2280 |
| C-C-O | 0.5198 | 0.3279 | 0.5216 | 0.5078 | 0.3335 | 0.3930 | 0.5327 | 0.3710 | 0.4021 | 0.2124 | 0.1277 | 0.3233 |
| C-C=C | 0.4657 | 0.2701 | 0.4430 | 0.2826 | 0.1815 | 0.3163 | 0.5047 | 0.2830 | 0.2706 | 0.2215 | 0.2267 | 0.1823 |
| C-C=N | 0.4441 | 0.4159 | 0.4376 | 0.3507 | 0.2075 | 0.3185 | 0.5171 | 0.3353 | 0.3205 | 0.2094 | 0.3175 | 0.2707 |
| C-N-C | 0.5209 | 0.3176 | 0.4586 | 0.3981 | 0.2915 | 0.4168 | 0.5378 | 0.4237 | 0.3597 | 0.1952 | 0.1475 | 0.1532 |
| C-N-N | 0.5889 | 0.2847 | 0.5403 | 0.4997 | 0.2626 | 0.4022 | 0.6605 | 0.4161 | 0.3505 | 0.2825 | 0.2702 | 0.1813 |
| C-N-O | 0.7019 | 0.4996 | 0.6338 | 0.6173 | 0.3263 | 0.4653 | 0.7070 | 0.5560 | 0.4943 | 0.3120 | 0.2002 | 0.3300 |
| C-N=C | 0.3646 | 0.4011 | 0.5133 | 0.3728 | 0.3105 | 0.3191 | 0.4433 | 0.4085 | 0.4517 | 0.3467 | 0.4092 | 0.3626 |
| C-N=N | 0.3597 | 0.6214 | 0.7639 | 0.7062 | 0.4400 | 0.4212 | 0.8326 | 0.7023 | 0.7380 | 0.3917 | 0.6954 | 0.7196 |
| C-O-C | 0.5111 | 0.4259 | 0.4872 | 0.4204 | 0.2865 | 0.3786 | 0.5478 | 0.4606 | 0.3765 | 0.1882 | 0.1113 | 0.3305 |
| C-O-N | 0.7893 | 0.3465 | 0.6637 | 0.6140 | 0.4312 | 0.5485 | 0.7520 | 0.6257 | 0.4988 | 0.4064 | 0.3336 | 0.4436 |
| C=C-N | 0.4580 | 0.4140 | 0.5415 | 0.3732 | 0.2359 | 0.3223 | 0.4075 | 0.3361 | 0.3226 | 0.2574 | 0.1760 | 0.2905 |
| C=C=C | 0.7593 | 0.6866 | 0.7793 | 0.7373 | 0.7445 | 0.7549 | 0.7752 | 0.7419 | 0.7603 | 0.7703 | 0.6388 | 0.8326 |
| N#C-C | 0.8151 | 0.2980 | 0.8326 | 0.5830 | 0.7437 | 0.7388 | 0.7928 | 0.3461 | 0.7120 | 0.7254 | 0.2252 | 0.7320 |
| N-C-N | 0.5157 | 0.3795 | 0.4179 | 0.5544 | 0.3058 | 0.4409 | 0.6764 | 0.4316 | 0.4464 | 0.2994 | 0.2604 | 0.2569 |
| N-C-O | 0.4713 | 0.2673 | 0.6054 | 0.5879 | 0.3926 | 0.4346 | 0.5923 | 0.4089 | 0.4987 | 0.3029 | 0.2612 | 0.4190 |
| N-C=N | 0.4598 | 0.3670 | 0.3450 | 0.3986 | 0.2175 | 0.3558 | 0.5498 | 0.2654 | 0.3531 | 0.2593 | 0.1366 | 0.1360 |
| N-C=O | 0.5275 | 0.3900 | 0.4285 | 0.2347 | 0.2664 | 0.3690 | 0.5719 | 0.3636 | 0.3695 | 0.1197 | 0.0378 | 0.2508 |
| N-N-O | 0.8326 | 0.6048 | 0.7791 | 0.7639 | 0.5862 | 0.5912 | 0.7447 | 0.7117 | 0.6875 | 0.4831 | 0.6708 | 0.5859 |
| N=C-N | 0.4598 | 0.3670 | 0.3450 | 0.3986 | 0.2175 | 0.3558 | 0.5498 | 0.2654 | 0.3531 | 0.2593 | 0.1366 | 0.1360 |
| O=C-N | 0.5275 | 0.3900 | 0.4285 | 0.2347 | 0.2664 | 0.3690 | 0.5719 | 0.3636 | 0.3695 | 0.1197 | 0.0378 | 0.2508 |

## E.2 LEAD OPTIMIZATION

We here give the details of interaction analysis. Since these tasks have provided a partial of the molecules as the context, it is hard to tell the superiority of generated substructures or geometries. Table. 21 and 22 gives details. it shows that

Table 21: Frequency of interaction type on lead optimization tasks.

| | Method | hydrophobic | hydrogen | water bridge | $\pi$-stacks | $\pi$-cation | halogen | metal |
|---|---|---|---|---|---|---|---|---|
| | REF. | 3.0000 | 3.0000 | 0.0000 | 0.0000 | 0.0000 | 0.0000 | 0.0000 |
| Linker | GRAPHBP | 5.5795 | 2.9812 | 0.0000 | 0.5362 | 0.3537 | 0.0000 | 0.0000 |
| | POCKET2MOL | 5.1879 | 3.2759 | 0.0000 | 0.1708 | 0.1082 | 0.0326 | 0.0000 |
| | TARGETDIFF | 5.3811 | 3.7133 | 0.0000 | 0.1675 | 0.0827 | 0.0404 | 0.0000 |
| | DIFFSBDD | 5.4132 | 3.1350 | 0.0000 | 0.1153 | 0.0132 | 0.0308 | 0.0000 |
| | DIFFBP | 7.0697 | 2.6017 | 0.0000 | 0.1226 | 0.0678 | 0.0204 | 0.0000 |
| Fragment | GRAPHBP | 3.0084 | 3.2071 | 0.0000 | 0.0947 | 0.0499 | 0.0196 | 0.0000 |
| | POCKET2MOL | 4.4064 | 4.0854 | 0.0000 | 0.1238 | 0.0956 | 0.0131 | 0.0000 |
| | TARGETDIFF | 4.3187 | 3.6436 | 0.0000 | 0.1040 | 0.0648 | 0.0405 | 0.0000 |
| | DIFFSBDD | 4.3086 | 3.6720 | 0.0000 | 0.1087 | 0.0089 | 0.0468 | 0.0000 |
| | DIFFBP | 5.6128 | 2.7838 | 0.0000 | 0.0552 | 0.0339 | 0.0175 | 0.0000 |
| Side Chain | GRAPHBP | 3.0173 | 1.2207 | 0.0000 | 0.2029 | 0.0321 | 0.0001 | 0.0000 |
| | POCKET2MOL | 3.0643 | 3.8911 | 0.0000 | 0.0872 | 0.0708 | 0.0194 | 0.0000 |
| | TARGETDIFF | 3.3468 | 3.6206 | 0.0000 | 0.0975 | 0.0527 | 0.0851 | 0.0000 |
| | DIFFSBDD | 3.6822 | 3.3479 | 0.0000 | 0.0271 | 0.0242 | 0.0098 | 0.0000 |
| | DIFFBP | 4.5444 | 2.5941 | 0.0000 | 0.0696 | 0.0349 | 0.0141 | 0.0000 |
| Scaffold | GRAPHBP | 0.9560 | 2.3953 | 0.0000 | 0.0000 | 0.0007 | 0.0007 | 0.0000 |
| | POCKET2MOL | 3.2220 | 3.4761 | 0.0000 | 0.0282 | 0.0295 | 0.0246 | 0.0000 |
| | TARGETDIFF | 3.5427 | 4.1047 | 0.0000 | 0.0612 | 0.0383 | 0.0303 | 0.0000 |
| | DIFFSBDD | 4.6225 | 3.2649 | 0.0000 | 0.0021 | 0.0138 | 0.1786 | 0.0000 |
| | DIFFBP | 5.0840 | 2.4424 | 0.0000 | 0.0061 | 0.0055 | 0.0110 | 0.0000 |

Table 22: Distibution of interaction type on lead optimization tasks.

| | Method | hydrophobic | hydrogen | water bridge | $\pi$-stacks | $\pi$-cation | halogen | metal |
|---|---|---|---|---|---|---|---|---|
| | REF. | 0.5000 | 0.5000 | 0.0000 | 0.0000 | 0.0000 | 0.0000 | 0.0000 |
| Linker | GRAPHBP | 0.5904 | 0.3154 | 0.0000 | 0.0567 | 0.0374 | 0.0000 | 0.0000 |
| | POCKET2MOL | 0.5912 | 0.3733 | 0.0000 | 0.0195 | 0.0123 | 0.0037 | 0.0000 |
| | TARGETDIFF | 0.5734 | 0.3957 | 0.0000 | 0.0178 | 0.0088 | 0.0043 | 0.0000 |
| | DIFFSBDD | 0.6216 | 0.3600 | 0.0000 | 0.0132 | 0.0015 | 0.0035 | 0.0000 |
| | DIFFBP | 4.1027 | 1.4830 | 0.0000 | 0.0213 | 0.0039 | 0.0021 | 0.0000 |
| Fragment | GRAPHBP | 0.4716 | 0.5027 | 0.0000 | 0.0148 | 0.0078 | 0.0031 | 0.0000 |
| | POCKET2MOL | 0.5051 | 0.4683 | 0.0000 | 0.0142 | 0.0110 | 0.0015 | 0.0000 |
| | TARGETDIFF | 0.5285 | 0.4459 | 0.0000 | 0.0127 | 0.0079 | 0.0050 | 0.0000 |
| | DIFFSBDD | 0.5289 | 0.4508 | 0.0000 | 0.0133 | 0.0011 | 0.0057 | 0.0000 |
| | DIFFBP | 0.6601 | 0.3274 | 0.0000 | 0.0065 | 0.0040 | 0.0021 | 0.0000 |
| Side Chain | GRAPHBP | 0.6746 | 0.2729 | 0.0000 | 0.0454 | 0.0072 | 0.0000 | 0.0000 |
| | POCKET2MOL | 0.4296 | 0.5455 | 0.0000 | 0.0122 | 0.0099 | 0.0027 | 0.0000 |
| | TARGETDIFF | 0.4647 | 0.5027 | 0.0000 | 0.0135 | 0.0073 | 0.0118 | 0.0000 |
| | DIFFSBDD | 0.5192 | 0.4721 | 0.0000 | 0.0038 | 0.0034 | 0.0013 | 0.0000 |
| | DIFFBP | 0.6262 | 0.3575 | 0.0000 | 0.0096 | 0.0048 | 0.0019 | 0.0000 |
| Scaffold | GRAPHBP | 0.2851 | 0.7144 | 0.0000 | 0.0000 | 0.0002 | 0.0002 | 0.0000 |
| | POCKET2MOL | 0.4752 | 0.5127 | 0.0000 | 0.0042 | 0.0043 | 0.0036 | 0.0000 |
| | TARGETDIFF | 0.4555 | 0.5278 | 0.0000 | 0.0079 | 0.0049 | 0.0039 | 0.0000 |
| | DIFFSBDD | 0.5719 | 0.4039 | 0.0000 | 0.0003 | 0.0017 | 0.0221 | 0.0000 |
| | DIFFBP | 0.6735 | 0.3235 | 0.0000 | 0.0008 | 0.0007 | 0.0015 | 0.0000 |

- The optimization on the side chains has the least effect on the interaction pattern in an overall result and has the greatest influence on the linker.

- GRAPHBP's failure in scaffold hopping also can be reflected by the fact that the interaction pattern cannot be produced with the molecules.

### E.3 CASE STUDY

**DRD3.** For DRD3, we give the distribution of different models' generated molecules in chemical space in Figure. 8 and 9. Different methods capture different clustering centers of the actives, while their distributions show a great gap from the GEOM-DRUG's randomly sampled molecules.

### E.4 VALIDITY

Validity is an important metric to evaluate whether the molecules generated by the models are valid. There are two methods to reconstruct the 3D positions of the atoms into a molecule with bonds, one is used in TARGERDIFF, POCKET2MOL and 3DSBDD, which we name as Refine; The second is to use Openbabel (Open Babel development team) the software, used in DIFFSBDD. However, using these methods to reconstruct molecules always carries the risk of broken bonds, which makes

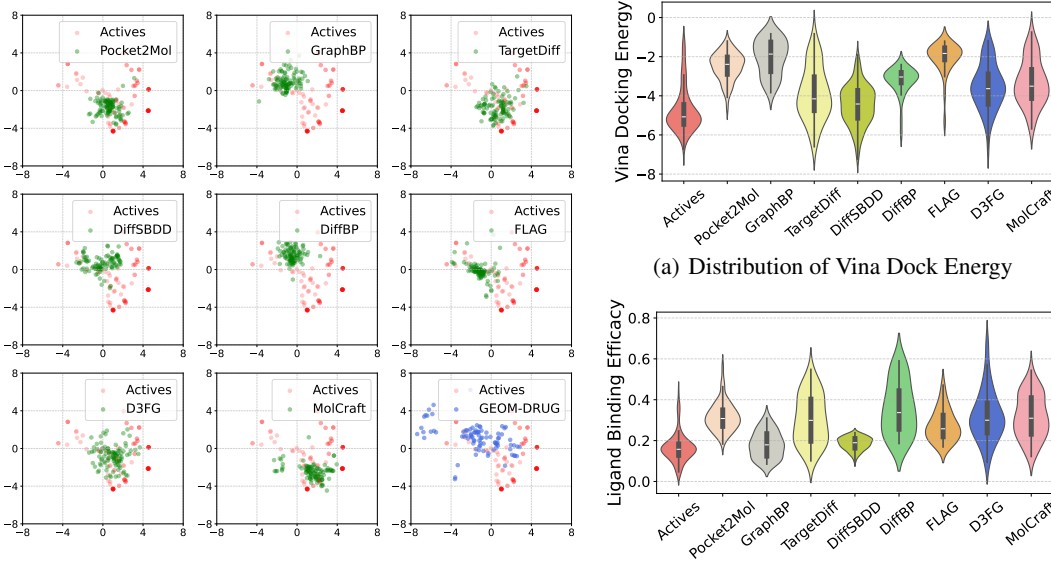

Figure 8: T-SNE visualization of chemical distributions of generated and active molecules on DRD3.

(a) Distribution of Vina Dock Energy

(b) Distribution of LBE

Figure 9: Binding affinities on DRD3.

Table 23: Denovo Validity and Rank

| Method | Validity | Rank |
|---|---|---|
| LiGAN | 0.42 | 12 |
| 3DSBDD | 0.54 | 11 |
| GraphBP | 0.66 | 10 |
| Pocket2Mol | 0.75 | 6 |
| TargetDiff | 0.96 | 1 |
| DiffSBDD | 0.71 | 7 |
| DiffBP | 0.78 | 4 |
| FLAG | 0.68 | 9 |
| D3FG | 0.77 | 5 |
| DecompDiff | 0.89 | 3 |
| MolCraft | 0.95 | 2 |
| VoxBind | 0.74 | 8 |

the strategy of selecting connected atoms to form fragments and ultimately the final molecule crucial. Here, we define a chemically valid molecule as one where the number of atoms in the largest fragment is greater than 85% of the total number of atoms, and we use this as the criterion for the reconstruction. Additionally, we employed a 'Refine + Openbabel' strategy, where if refinement is unsuccessful, Openbabel is added as a reconstruction method. Under this definition, the validity of various methods in the *de novo task* is shown in Table. 23. DecompDiff and TargetDiff have the highest validity.

