# OpenReview forum: "CBGBench: Fill in the Blank of Protein-Molecule Complex Binding Graph"
_ICLR.cc/2025/Conference — ICLR 2025 Spotlight_

### Official Review · Reviewer_ftZU · 2024-11-04

**Soundness:** 3
**Presentation:** 3
**Contribution:** 3
**Rating:** 8
**Confidence:** 4

**Summary:**

The paper introduces a new benchmark for generative models in structure-based drug design. Recently, many machine learning models have been proposed to help solve the problem of designing compounds that can interact with their protein targets. However, the comparison of these methods is difficult due to the lack of standards and relevant benchmarks. CBGBench is a benchmark that aims at standardizing model comparison and proposing evaluation metrics that can be used to assess the performance of generative models. Additionally, the benchmark contains tasks that can be used to evaluate models' capabilities of fragment linking, scaffold hopping, fragment growing, and side chain generation. Several recent generative models were compared using the proposed benchmark, and conclusions were drawn from these experiments. Moreover, a case study was conducted to demonstrate the quality of compounds generated by these methods for two selected GPCR targets.

**Strengths:**

Originality:
- A new benchmark is proposed for comparing the performance of recently developed SBDD methods.
- This benchmark also includes additional tasks like fragment linking or scaffold hopping, for which do not exist any standardized benchmarks.
- A taxonomy of SBDD generative models is proposed.

Quality:
- The benchmark covers a wide range of generative models, including autoregressive and one-shot generative models.
- Both evaluation methods and scoring network architectures are standardized.

Clarity:
- The benchmark is clearly defined, and the diagrams in Figures 2 and 3 explain different tasks and model types.
- The evaluation metrics are explained in an easy-to-follow way.

Significance:
- The proposed benchmark should accelerate research on new generative models that generate molecules binding to a given protein.
- The benchmarking code is attached to this submission, making it easier to test new models in the future.

**Weaknesses:**

Originality:
- This work does not discuss other benchmarks for generative methods. In particular, Zheng et al. [1] have recently introduced an SBDD benchmark for generative models.

[1] Zheng, Kangyu, et al. "Structure-based Drug Design Benchmark: Do 3D Methods Really Dominate?." arXiv preprint arXiv:2406.03403 (2024).

Quality:
- The requirement that "each connecting fragment must consist of more than five atoms" seems arbitrary. Why are models that occasionally generate fewer atoms considered insufficient?
- The benchmark could propose a method of computing confidence intervals, e.g. by sampling the generated molecules. It would facilitate the judgment of whether one of the tested models is significantly better than the others.

Clarity:
- "Real-world targets" may be a confusing name for the structures used in the case study. The protein structures in the CrossDocked2020 dataset were also real-world targets, which might confuse some readers. Maybe it would be better to call the structures in the case study as selected targets.
- Additionally, I recommend rephrasing statements like "the established evaluation protocols exhibit strong consistency and generalizability on real-world target data." The real-world target data could be understood as binding data from laboratory experiments, but docking experiments usually do not correlate with such experimental data.
- The text in Figures 4 and 5 may be too small.

Minor comments:
- A typo in line 68: "Vina enery"
- A typo in line 475 "the neurotrans-625 mitter"

**Questions:**

1. What is the difference between fragment growing and side chain decoration on the implementation level? The text mentions ligand decomposition for the fragment growing task, but the detailed procedure of the decomposition is not explained. In Figure 3, how is the scaffold selected in the case of side chain decoration?
2. Some of the employed evaluation metrics compare the generated distribution of molecules to a known distribution of molecules. Would this approach not penalize methods that generate molecules with novel scaffolds? Also in the t-SNE plot (Figure 4) that compares the set of generated molecules to the actives, the compounds generated by a model can be significantly different from the known actives but still active. A good example of this behavior would be genetic algorithms that optimize docking scores. These methods are not trained on any data, so they can easily learn to produce compounds that have a unique structure and dock well to the target protein.

---

> ### Author Response · Authors · 2024-11-14
> **Thanks! Response to Weakness**
>
> Thanks for your recognition of our contribution, and here we will answer your questions to eliminate your concerns one by one:
>
> **Reply to Originality:**
>
> We have carefully reviewed the article by Zheng et al. [1] and found that it was published on June 4, 2024, contemporaneously with our work. Thus, we believe that concerns regarding the novelty of our research are unfounded. Additionally, we would like to highlight several distinctions and advantages of our study in comparison to this work:
>
>  - 1. Zheng et al. primarily aim to demonstrate that 1D/2D molecular generation methods remain highly competitive compared to 3D approaches. In contrast, our work focuses solely on generative 3D SBDD tasks. Moreover, we provide a more comprehensive evaluation of 3D molecular generation methods, incorporating over 10 3D-based methods compared to the five included in Zheng et al.
>  - 2. In terms of evaluation metrics, beyond Vina Dock Energy, SA, and QED for drug properties, we include additional assessments such as PLIP interaction analysis, clash analysis, and geometric fidelity. Our 3D-based comparisons are therefore broader and more detailed.
>  - 3. We have incorporated four additional lead optimization tasks, including linker, fragment, scaffold, and side-chain design. These aspects are rarely covered in previous benchmark or SBDD task-oriented papers.
>  - 4. Our work unifies the code framework, whereas Zheng et al.’s published GitHub repository does not include training or generation code for the methods discussed. This difference implies a significantly greater engineering effort on our part, potentially making a substantial contribution to the community. Please refer to the '.zip' file in the supplementary material for further details.
>  - 5. We conducted experiments on real disease targets, ADRB1 and DRD3, which are rarely explored in previous benchmarking and SBDD task-oriented studies.
>
> Therefore, we believe that (1) questioning the novelty of a contemporaneous study is not entirely fair in this context; and (2) CBGBench possesses distinct differences and advantages compared to the work you referenced. Besides, since Zheng et al. also gives lots of insights in SBDD tasks, we have added it as a reference in the newly updated Line. 539 for the reader to understand the field easily.
>
> **Reply to Quality:**
>
> - We choose 5 as the threshold for considering the group of atoms because too few atoms cannot form appropriate fragments. Criteria of `5 atoms' primarily references the atomic number constraints used in the DeLinker[2], where the design likely focuses on common pharmacophoric substructures like five-membered rings for fragment truncation. In practice, you can freely define these based on your own requirements, and the relevant code has been uploaded.
> - Thanks for you advice on the confidence intervals computing method. We think it is really valuable, since in the macro-biomolecule design, the task of `quality identification' is very important, to tell the biologist whether the generated biomolecules are reliable. We will take the advice as future work, to establish an SBDD-version generation quality identification model, to facilitate the judgment of whether one of the tested models is significantly better than the others.
>
> **Reply to Clarity:**
>
> - We aggree with you that the phrase is confusing, so we change it into disease target, since **DRD3** is a type of dopamine receptor primarily found in the brain, related to **Schizophrenia and Parkinson’s disease** and **ADRB1** is a β1-type adrenergic receptor related to **hypertension, heart failure and Arrhythmias.** In comparison, in CrossDocked, the protein pockets are derived from structures in PDBbind, forming protein-ligand complexes that exist in the real world. However, while these receptors are indeed real-world entities, they are not necessarily disease targets. Therefore, in the newly uploaded version, we have revised references to “real-world target” to “real-world disease target” or “real-world target related to disease” to avoid any potential ambiguity.
> - We have incorporated your suggestion and revised the text to: “the established evaluation protocols exhibit consistency and generalizability on real-world target data in evaluating the Vina Dock Energy.”
> - We have changed the font size in the Figure in the latest updated version.

---

> > ### Author Response · Authors · 2024-11-14
> > **Response to Questions**
> >
> > **Reply to Questions:**
> >
> > 1. Fragment growing typically involves a larger functional group, whereas side-chain decoration may involve multiple tiny fragments, such as one scaffold corresponding to five tiny fragments[3]. In fragment growing, we choose to cleave non-cyclic single bonds to obtain two fragments, which should satisfy the two mentioned rules. Figure 3 is merely to illustrate the concept of a scaffold. In the selection of scaffolds, we adopt the BM scaffold definition, meaning the scaffold includes all cyclic structures within the molecule.” The detailed implementation can obtained through Line 314 as the definition of scaffold_decomp function in the file of  `./CBGBench-master/repo/datasets/parsers/molecule_parser.py`
> > 2. Firstly, in lead optimization, the goal is to generate molecules structurally distinct from the original, so we have intentionally avoided restrictive metrics like substructure distribution that could limit molecular diversity. Instead, we focus on comparisons based on Vina Energy and drug properties, which do not penalize methods that generate molecules with novel scaffolds. Secondly, in the t-SNE analysis, we followed the approach in Figure 1 of [4] to examine whether the distributions of generated molecules are chemically consistent with those of the actives. We do not consider methods with diverse distributions to be inferior; the visualization serves as an exploration tool rather than a ranking criterion. Additionally, we agree with your observation that some optimization methods can generate molecules with diversity and activity. In addition to genetic approaches, iterative target optimization (similar to genetic methods but not training-free) can also enhance molecular activity [5]. We intend to explore such methods in future work related to molecular optimization.
> >
> > [1]  Zheng, Kangyu, et al. "Structure-based Drug Design Benchmark: Do 3D Methods Really Dominate?." arXiv preprint arXiv:2406.03403 (2024).
> >
> > [2]  Fergus Imrie et al. Deep Generative Models for Linker Design, https://pubs.acs.org/doi/10.1021/acs.jcim.9b01120
> >
> > [3]  Zhang et al. Deep Lead Optimization: Leveraging Generative AI for Structural Modification. https://pubs.acs.org/doi/10.1021/jacs.4c11686
> >
> > [4] Mingyang Wang et al. A deep generative model for structure-based de novo drug design https://pubs.acs.org/doi/10.1021/acs.jmedchem.2c00732
> >
> > [5] Zhou et al. DecompOpt: Controllable and Decomposed Diffusion Models for Structure-based Molecular Optimization, https://arxiv.org/abs/2403.13829

---

> > > ### Comment · Reviewer_ftZU · 2024-11-21
> > > **Thank you for the response!**
> > >
> > > Thank you for your response and clarifications. My concerns have been resolved.
> > >
> > > I would like to clarify that my comment requesting more discussion about other benchmarks, such as those by Zheng et al., was not meant to undermine the novelty of the proposed benchmark. I believe that a brief comparison of CBGBench with other benchmarks, highlighting key differences and the potential advantages of CBGBench, would offer important context for readers and emphasize the originality of your work.
> > >
> > > Regarding the confidence interval computation, I am thinking that maybe one can compute many bootstrap samples of the generated molecules and calculate evaluation metrics on these samples. Then 2.5th and 97.5th percentiles could be used as a 95% bootstrap confidence interval. What do you think about this approach? Probably this method requires more consideration to make this process deterministic (avoid variance due to random sampling).

---

> > > > ### Author Response · Authors · 2024-11-22
> > > > **Thanks for your advices.**
> > > >
> > > > Thank you for your response. We apologize for misunderstanding your feedback. We indeed recognize the value of similar benchmarks. However, due to space limitations, it is difficult to add additional discussion in the main text. As a result, we have included a brief discussion in Section 6 Limitations and Conclusion (Line 538). Additionally, we have added a comparison between our work and the mentioned benchmark in Appendix A (Line 938 - 942), highlighting their respective strengths and differences.
> > > >
> > > > Besides, we agree with you that using bootstrap methods to take some confidence intervals into consideration is reasonable. Besides, for the mean value, we think a large bootstrap number will approximate the raw average reported mean metrics, or the difference is very minor, so we think it might be unnecessary to report the statistical mean obtained from bootstrap again. Besides, as the numerical values in the comparison table are already quite dense, reporting the confidence intervals again might be overwhelming for readers and could reduce the readability of the article. However, we still highly value your feedback. Therefore, we have added the CI calculation in the evaluation code of the benchmark. You can find it in the updated supplementary.zip file under `CBGBench-master/evaluate_scripts/cal_chem_results.py` at lines 9–14 and lines 115–146. We sincerely thank you for your valuable feedback and insightful comments.
> > > >
> > > > Finally, we would like to know if we have addressed all your concerns at this stage. If you have any further related concerns, please feel free to share them with us, and we will be more than happy to address them. If you are satisfied with our responses, we kindly ask if you would consider reevaluating the manuscript and providing a stronger recommendation. We believe this work, along with the accompanying code base, has the potential to make a significant impact on the community and drive its development forward. If you share this view, your stronger recommendation would be a great motivation for us to continue contributing to the SBDD field. Thank you very much!

---

> > > > ### Author Response · Authors · 2024-11-25
> > > > **Many thanks.**
> > > >
> > > > Dear Reviewer ftZU:
> > > >
> > > > Thank you for your insightful and helpful comments once again. We suppose we have thoroughly considered all the feedback and provided a point-to-point response to you.
> > > >
> > > > The deadline for author-reviewer discussions is approaching. If you still need any clarification or have any other questions, please do not hesitate to let us know.  We hope that our response addresses your concerns to your satisfaction and we would like to kindly request a reconsideration of a higher rating if it is possible.
> > > >
> > > > Best regards,
> > > >
> > > > Authors.

---

> ### Author Response · Authors · 2024-11-28
> **Request with sincere thanks.**
>
> Dear Reviewer ftZU:
>
> Thank you very much for recognizing our work. We have provided responses to the fragment growing and other molecule splitting rules you raised. Additionally, the revised manuscript we uploaded has incorporated your suggestions, including: replacing the less appropriate term “real-world target” with “real-world disease target”; citing and discussing the references you suggested (Appendix Line937 - 943); and updating the newly uploaded code version to include the bootstrap method for confidence interval estimation as per your recommendations.
>
> Given the extended discussion period, we sincerely ask if there are any unresolved concerns or whether our current responses have addressed all your doubts about the manuscript. Considering that all reviewers with positive opinions have provided satisfactory feedback and stronger recommendations after our responses, we hope that if you have any remaining questions, you feel free to raise them.
>
> We sincerely look forward to your response.
>
> Best regards,
>
> Authors.

---

> > ### Comment · Reviewer_ftZU · 2024-12-01
> >
> > Thank you for your response and for incorporating my suggestions. I do not have any further questions. I raised my score to 8.

---

> > > ### Author Response · Authors · 2024-12-02
> > > **Many Thanks.**
> > >
> > > We sincerely appreciate your valuable feedback.  Happy Thanksgiving!

---

### Official Review · Reviewer_G2c6 · 2024-11-05

**Soundness:** 3
**Presentation:** 2
**Contribution:** 2
**Rating:** 5
**Confidence:** 2

**Summary:**

The paper proposes an extended benchmark protocol for SBDD methods, and does a large comparison of existing literature, and draws some insights from the results.

-----

Post-response update. My initial confusions have been adressed in the response, as well as issues in motivation. This is a useful contribution to SBDD's, with weaknesses wrt incrementality, presentation and experimental insights. I'm increasing my score to borderline, and I'm still leaning on rejection while I wouldn't mind acceptance.

**Strengths:**

- The paper does a comprehensive benchmark, with lots of interesting results.
- The authors contribute a unified codebase, which could be very useful for future research.

**Weaknesses:**

- The motivation of the work is weak. I’m not sure why we need all of this stuff, or what questions the paper answers. The paper seems to establish an extended benchmark protocol that adds few extra evaluations to standard benchmarks. This could be useful, but I found the new additions to be a bit hand-wavy. The paper also proposes “unified” tasks and notation, but why? These tasks and notation is already common.
- The novel contributions of the work seem limited. The data and tasks seem to be from earlier works, and the benchmarks are added with just few extra evaluations, which also all seem to already exist. The main contribution is collecting all of this into one place. The performance comparison is a good contribution, but similar comparisons are done in every paper. The proposed new “unified” notation or task framing seems all well known. The main new contribution is probably the unified codebase, which can be very useful. The paper draws a bunch of conclusions from the results, which are interesting, but it’s unclear how novel they are.
- As a benchmark paper the presentation of the results is limited: there are too many decimals, no standard deviations, and there are lots of massive tables of “raw” performance values, which are not digested into transferable insights. The analysis is superficial, and there are no ablations or attempts to explain how or why the different methods perform differently.
- As a survey paper I didn't find this paper particularly good at summarising or organising the domain in terms of methods, data or metrics. Use of math is inconsistent.

**Questions:**

- It would be useful to extend the method table by also including which tasks they support, and their architectures and losses
- Training losses should be discussed
- The paper first describes the main de novo task very superficially, then it describes the subtasks more in-depth, then it goes back to describing the de novo task more in detail, and finally goes back to subtasks. It would have been much clearer to separate the main/sub tasks into different parts of the paper.
- The task description is missing the objectives/metrics. Overall I didn’t really understand the role of the four subtasks. These are very specific, and surely each method needs to have custom support for them. Which ones do? Do we expect new methods to support these 4 tasks? Isn’t it enough that a method is good at the “basic” de novo generation? I’m a bit confused why we care about these at all, or what is their significance? It’s also unclear how they relate to the probabilistic model or ML modelling. Are these some kind of conditionals?
- It’s unclear if the 4 tasks in table 2 have something to do with the main task. That is, do they share data in some sense? Are the 4 tasks more fine-grained versions of the de-novo data?
- I’m a bit confused what’s the point of the “unified” probabilistic model notation. It seems that this is used nowhere in the paper after it’s introduction.
- Why would you look at the MAE between two distributions? This doesn’t seem sensible. The distribution sizes and domains and types are mostly undefined [please use precise math]
- What does it mean to measure “atom types” of functional groups? I don’t understand. I though that you compared just some distributions of 25 groups, which should give you a probability vector of length 25 to be compared. The at/rt/fg should have no role here. [It would help so much to define the stuff in precise math so that I wouldn’t have to guess.]
- I fail to see where the accuracy of the de novo generation is described. I thought the task is to predict the correct ligand for a known protein (assuming there is only one correct ligand). Surely you want to measure how often you get this right. The paper also talks about probabilistic model of p(M|P). Where is this density evaluated: shouldn’t we see some logp values in the results?
- Or is the task to just generate some ligands (irrespective if it was the correct one), as long as they have good binding and properties, or come from some distribution? If the task really is generative, then one expects to use learning target of maximizing the model likelihood of the observed data (both training and test folds). I don’t see this in the paper either. The learning problem needs to be precise.
- Overall it seems that some metrics in the paper are about checking that the summary statistics of $p(ligand)$ or $p(ligand|protein)$ match the true distributions, and some are about making sure some $fitness(ligand,protein)$ is high. This is not formalised well, and it leads to the metrics and learning setting being vague and hand-wavy.
- I fail to see the motivation for the different Vina scores. If Vina is pathological, then why would we still use it? I don’t see how IMP or MPBG helps either if they are based on Vina. I don’t see convincing arguments why the LBE fixes the Vina pathology: the table 11 has values all of the place. I think here less would be more, and it would be much better to just have one good metric for binding than lots of binding metrics of varying quality.
- I did not understand the PLIP stuff.
- Overall the metrics are difficult to interpret. I don’t really know what the numbers mean. Is 0.4382 MAE good or bad? Is a 0.2345 JSD high or low? Is Vina -3.75 good or bad? Is IMP 22 good? I have no idea. It would be insightful to visualise the distributions, or use some human-understandable metrics. For instance, you could use distribution overlap percentage or something else.
- It would have been useful to show some example generations, and their corresponding metrics.
- Using 5M iterates in each method seems unfair. Different methods train in different ways. I think you need to analyse the discrepancy between your results and the published results to clarify this.
- It seems that you change some methods architectures for no reason. This is not fair and will nullify the corresponding results. You can’t claim to benchmark method X if you change the method X from the publication.
- It’s unclear if you reimplement the methods in your codebase, or just collect published codebases in one place. Can you clarify?
- “We generate 100 molecules per pocket in test”. How do you do this? Why do you do this? Didn’t we have test data that tells us this? Now there are 10000 test molecules, while table 2 says that there are only 100. The learning setting is confusing, and using math to describe things would help. It seems that the entire setting in this paper is some kind of $D[p_gen || p_obs]$, which is not properly formalised.
- “We show the interaction analysis and chemical property in the main context and omit the interaction pattern analysis since maintaining the patterns is not necessary for lead optimization.” I don’t understand this. How do you use “chemical property” as context? How do you use “analysis” as context? What is "context"?
- I don’t understand what you do in the subtask training. Surely a method can’t be just applied in eg. linker generation if it wasn’t designed to do it? Are all 6 methods ones that support it, or do you somehow kludge this support to them? I’m really confused what do you train for extra 1M steps. What are even the loss functions? This is all super vague.
- The results of subtasks are all over the place. I’m not sure what can you conclude from this.
- Using ECFP to compare molecules in sec 5.3. is a very weak approach, and running it through tsne2d makes it even worse. This analysis has little to no value.
- The setting in 5.3. is weak. Thera are only 100 random controls (why not way more, like a million?). It’s unclear how many molecules you sample, but this should be a large number.
- Sec 5 concludes that the evaluation protocols are very consistent with real-world behavior. By far the best method in sec 5 is the DiffSBDD, which has by far best LBE, and ok Vina’s (Vina is pathological, so we should look at LBE anyways). But in Table 7 we see DiffSBDD being ranked 10/12, ie. one of the worst methods. It’s pretty clear that the real-world performance and de novo benchmark performance is not consistent.

---

> ### Author Response · Authors · 2024-11-15
> **Oveall Response**
>
> Given the numerous issues raised in this review, we have broken down our responses into several sections to address each point individually. Additionally, since your questions were not numbered, we have re-labeled them for clarity. Due to space limitations, we did not copy the re-labeled questions with number, and you may re-number your questions and cross-reference them accordingly.  The questions and weaknesses can be classified into the following parts:
>
> ### I. Novelty& Contribution:
> **Weakness 1, 2**
>
> ### II. Presentation:
>
> **Weakness 3, 4**
>
> **Question 1, 2, 3, 4, 6, 20, 22, 23 25**
>
> ### III. Metrics:
>
> **Question 7, 8, 9&10, 11, 12, 13, 14**
>
> ### IV. Data:
>
> **Question 5**
>
> ### V. Training and Sampling Protocol:
>
> **Question 15, 16, 17, 18, 19, 21, 24**
>
> ***`Besides, we friendly recommend the reviewer to get more familiar with some representative prior works, such as TargetDiff[1] and Pocket2Mol[2], to avoid questions that challenge well-established concepts and common consensus in the field. :)`***

---

> > ### Comment · Reviewer_G2c6 · 2024-11-15
> >
> > Thanks for the response, which resolved many of my concerns.
> >
> > I think the discussion here revolves around whether a manuscript needs to be self-contained. I don’t think it’s a requirement, but does increase paper’s accessibility and thus impact. For a benchmark paper that sets up new evaluation protocols having transparency and precision is especially important. The paper is light on math, which necessitates going back to other papers to look up definitions of some parts of the paper. Writing out the math of the entire modelling pipeline to the appendix is highly recommended.

---

> > > ### Author Response · Authors · 2024-11-15
> > > **Response**
> > >
> > > Firstly, we do not believe that our paper lacks self-containedness. In fact, you are the only reviewer among the five to raise concerns related to presentation and self-containedness. We kindly suggest that you conduct a more thorough evaluation after gaining a deeper understanding of prior works, conducting further research and reviewing our responses one by one to determine whether they help you better understand the raised issues. As this paper focuses on a highly specialized topic rather than serving as an introductory review, it may be challenging to fully grasp the issues and conclusions without sufficient background knowledge.
> > >
> > > Moreover, SBDD is a field that inherently requires substantial background knowledge. For readers who are less familiar with molecular structure generation, 3D de novo molecular design, or protein structure design, it can indeed be challenging to quickly define the relevant problems and provide a comprehensive overview within the constraints of a 10-page article. If mathematical formalism and detailed introductions to this field are needed, there are already some relevant works, such as [1]; And for [2] it is more introductory in nature, making it suitable for professionals in both biology and computer science-related fields. We recommend that readers refer to these reviews to gain foundational knowledge of the field, and then use the conclusions of our benchmark to derive deeper insights. “Writing out the entire modeling pipeline in mathematical detail”, as suggested, would require a review-length article spanning dozens of pages. To address this, we will add citations to introductory works to guide beginners in understanding the field more efficiently. Besides, we have updated the training objectives of the models to help to summarize the probabilistic models each model uses.
> > >
> > > Given that we have addressed most of your concerns, we kindly ask that you clearly enumerate any unresolved issues or additional concerns. At present, all feedback we have received has been constructive, and reviewers with deeper familiarity in this field have promoted a positive recommendation and already updated their responses with stronger recommendations. In light of this, we sincerely hope that after reviewing our responses to all the questions raised and gaining an understanding of some representative works, you can reassess the novelty, contribution, and potential impact of this paper. Thank you.
> > >
> > > [1] Zhang Zaixi et al. Geometric Deep Learning for Structure-Based Drug Design: A Survey. https://arxiv.org/pdf/2306.11768
> > >
> > > [2] Camille Bilodeau, Generative models for molecular discovery: Recent advances and challenges, https://wires.onlinelibrary.wiley.com/doi/10.1002/wcms.1608

---

> ### Author Response · Authors · 2024-11-15
> **I. Reply to Novelty&Contribution**
>
> **Weakness 1:**
>
> The motivation is clearly stated in both the abstract and introduction. If there are any specific concerns, we welcome precise feedback. While **de novo** tasks are indeed common, lead optimization tasks have received far less attention in the literature.
>
> **Weakness 2:**
>
> Our contributions include a unified setting of tasks, a unified codebase, a unified evaluation protocol, extensive empirical study and insights derived from experimental results. The framing of tasks is not well-known; if it is, please provide specific examples. Simply asserting a lack of novelty is easy, but pointing out similar prior work would constitute a more solid argument.

---

> ### Author Response · Authors · 2024-11-15
> **II. Reply to Presentation**
>
> **Weakness 3:**
>
> The use of additional decimal places is because many values cannot be meaningfully distinguished using only two decimal points, so we have adopted four decimal places for precision. The absence of standard deviations is due to the generative nature of the task, where the comparison focuses on the differences between distributions. Sampling 10,000 molecules for distribution comparison is an established protocol followed by nearly all prior studies, and we believe that adhering to a reasonable protocol based on published works is appropriate.
>
> The metrics we present are those typically reported in prior studies, and the benchmarking tables follow the format in [3]. If a ranking is all that’s needed, please refer to the last column in each table. In fact, the performance differences among these methods are minimal, and comparing them does indirectly serve as a form of ablation analysis. We have clearly outlined the reasons for the performance differences in the Conclusion section, so please review this in detail. Additionally, we would appreciate specific examples to clarify what kinds of insights are being sought.
>
> **Weakness 4:**
>
> We would appreciate it if specific evidence for the allegations could be provided. If the following questions are specific concerns, we would be happy to address them one by one.
>
> **Question 1 & 2:**
>
> We have added an additional table in the Appendix to clarify the supported tasks in Table 9, along with the corresponding losses in Table. 10. Besides, the architecture has already been outlined in Table 9.
>
> **Question 3:**
>
> We aim to introduce the task of **de novo Complete Binding Graph** to unify the lead optimization process. Subsequently, in the experiments, we first compare the performance on **de novo design** and then evaluate the related performance on **lead optimization**. Our paper's presentation follows a structured flow: `introduction of tasks and corresponding data -> presentation of evaluation protocols -> experiments`.
>
> **Question 4:**
>
> Many methods require modifications to be adapted for these tasks. Although the training objectives are usually the same, the conditions in conditional probabilistic models differ, necessitating various adaptations, such as zero-centering-of-molecule techniques. Of course, we would prefer there will be newly established models better suited to these tasks; however, as a benchmarking study, we primarily utilize existing models and adapt them for relevant tasks instead of proposing a new one. Moreover, models designed for **de novo** generation are not always effective in **lead optimization**. A typical example of this is autoregressive models, which may perform well in de novo generation but fall short in lead optimization.
>
> **Question 6:**
>
> The use of this notation is to unify the models under the task of “Fill-in-the-gap in Complex Binding Graph.” If a unified notation system were to be applied to describe all models, the article would become overly verbose and lean more toward a review-style paper rather than a benchmark and dataset study. Please differentiate the purposes and writing styles of these two types of articles. For reference, you may look at Atom3D [4].
>
> **Question 20:**
>
> The context means the text in the main body, we here change the discription as main text to avoid confusion.
>
> **Question 22:**
>
> Please refer to Line 452 - 459 to figure out if you are sure that the conclusion is right or wrong. We here need more detailed problems.
>
> **Question 23:**
>
> The visualization is a very common technique in the field of Computational Chemistry, for example [5] and [6]. Please refer to them.
>
> **Question 25:**
>
> Although Vina is known to have limitations, there is currently no better computational method widely available to replace docking as an initial screening metric. Therefore, we have continued to use it. From the results in Table 7, it is evident that DiffSBDD underperforms. Furthermore, in the real-world target case study, DiffSBDD also appears to perform poorly, which demonstrates the consistency of these two conclusions.
>
> We kindly ask the reviewer to carefully read our paper to fulfill the responsibilities of peer review. The inconsistency between 'A' and 'B' that you pointed out does not pertain to the same objects we described, and thus the claim of inconsistency does not apply.

---

> ### Author Response · Authors · 2024-11-15
> **III. Reply to Metrics**
>
> **Question 7:**
>
> MAE is used to compared point to point. For example, for a drug dataset which can bind to certain pocket, where the average molecule contains 2 benzene rings and 4 hydroxyl groups, its distribution can be represented as  $\text{Multinomial}(1/3, 2/3)$. If, in the generated molecule set, each molecule contains 1 benzene ring and 2 hydroxyl groups, the overall distribution matches, and the JS divergence would be optimal. However, for such molecules, their pharmacophore, which interacts with the drug, may require 2 benzene rings and 4 hydroxyl groups. In this case, KL divergence fails to capture the differences in functional group distributions. To address this limitation, we introduce  $\text{MAE} = 0.5 \times (|2 - 1| + |4 - 2|)$  as a point-to-point comparison metric, which is reasonable and aligns with the approach in D3FG [7].
>
> For other metrics, a similar rationale can be applied. If you find any metrics difficult to understand, please let us know, and we are extremely pleased to provide further examples to clarify the metrics you are struggling with. :)
>
> **Question 8:**
>
> ‘Atom type’ is denoted by AT, 'Functional Group' is denoted by FG and RT means 'Ring type'.  They are three different and independent evaluation frameworks. For atom type, we only consider the distribution of seven atom types: C, N, O, F, Cl, P, and S (Table 15 in the latest-uploaded version). This results in a generated probability distribution  $\hat{p} \in \mathbb{R}^7$  with  $\sum_i^7 p_i = 1$ , which can be compared with the true distribution  $p \in \mathbb{R}^7$  to compute the divergence. For functional groups (FG),  $p$  is a 25-dimensional probability vector (Table 17 in the latest-uploaded version, where we show the top 10 dimensions). For rings,  $p$  is a 6-dimensional vector (Table 16 in the latest-uploaded version).
>
> **Question 9&10:**
>
> The SBDD task is not about generating the “correct” ligand; the task aims to explore the chemical space of ligands that can bind effectively to the target. The requirement is for the generated ligands to exhibit properties such as realistic molecular geometry, drug-like properties, pharmacophoric substructures, expected interaction types, and the ability to bind well to the target protein. From this perspective, we suggest reevaluating the protocol and metrics. “Correctness” has never been the focus of this field—if your model could only predict existing, validated drugs, the goal of “drug discovery” would no longer exist.
> Additionally, given that many probabilistic models cannot compute precise likelihoods, such as MolCraft using BFN and LiGAN using VAE, comparing the NLL (negative log-likelihood) on the test set becomes challenging to implement.
>
> **Question 11:**
>
> Section 4 provides clear comparisons of the metrics and explicitly explains which metrics are better when larger and which are better when smaller. For metrics like JSD, which match the true distribution, the goal is to minimize it, and this does not create any conflict. Additionally, each metric is ranked internally, so differences in metric scales do not affect the comparisons. Could you please specify which aspects made you feel that the comparisons were vague? This would help us address your concerns more effectively.
>
> **Question 12:**
>
> The three Vina Energy represents the energy calculated with intial pose, optimized pose and optimized conformations.  Although the predictions made by Vina are neither fully precise nor comprehensive, truly accurate validation requires wet-lab experiments, which are neither feasible nor affordable for screening generated molecules. Furthermore, currently, no widely recognized tools superior to Vina have been adopted in the field (as evidenced by the references we cited). Therefore, our aim is to build upon Vina and address some of its limitations.
>
> Regarding LBE, we suggest reviewing the analyses provided in Appendix B2 and B4. If you still have concerns, we welcome further discussion. Additionally, for the presentation of metrics, you may refer to the main experimental tables in previous works such as [1].
>
> **Question 13:**
>
> Please refer to Sec.4 Line 255 - 275, and besides, PLIP tool is commonly used to analyze the interaction pattern (Figure 2c in page 1024 in [8] and Figure 3c in [9]). If you are interested in PLIP tool, we here give you a detailed paper for reference [10][11]. We welcome further discussion.

---

> > ### Author Response · Authors · 2024-11-15
> > **III. Reply to Metrics-2**
> >
> > **Question 14:**
> >
> > We provide baseline metrics as references molecules for some metrics, such as IMP% meaning that how much do the generated molecules perform better than the reference ones in binding energy, and you can refer to Sec.4 and Sec. 5 to have an intuition of the metrics. Regarding other metrics without obvious baselines, it is well-known that these metrics are primarily used for comparisons between different methods, and some of the metrics are not so human-understandable. It is often difficult to determine whether a specific value of a metric is inherently good or bad. For example, is an FID of 2.20 or an inception score of 9.90 good or bad in the context of generative model comparisons? Why is it considered good or bad? Similarly, for tasks of varying difficulty, even with straightforward metrics like accuracy, it is challenging to derive a clear sense of quality from the values alone. For instance, if a model achieves an accuracy of 0.9 on CIFAR100 but 0.85 on ImageNet1000, is the model performing well or poorly? Metrics are meant for comparison; many of them are not inherently human-understandable.
> >
> > Furthermore, these metrics are commonly used in prior SBDD papers that I have listed. I kindly and respectfully suggest that you take the time to understand the calculation of these metrics and the purpose of performing this task in this field.

---

> ### Author Response · Authors · 2024-11-15
> **IV. Reply to Data and V. Training and Sampling Protocol**
>
> ### **IV. Reply to Data**
> **Question 5:**
>
> The dataset are all constructed based on CrossDocked2020. The four task aims to generate different part of a molecule, so the preprocessing of the fragmentation of the molecules in CrossDocked2020 is different, leading to different 4 subtasks and the corresponding 4 datasets. You can refer to Sec. 3 for details.
>
> ### **V. Reply to Training and Sampling Protocol**
>
> **Question 15:**
>
> In fact, every paper presents various examples highlighting its strengths and weaknesses. We plan to include generated molecular case studies in the appendix in future versions to provide more intuitive visualizations.
>
> **Question 16:**
>
> In fact, we reviewed the standard training protocols currently in use and found that 5M iterations represent the maximum used. Therefore, we standardized on 5M iterations for all models to ensure fairness, avoiding cases where some models might fail to converge with fewer iterations. The final model is selected based on the lowest validation loss, which we believe makes this unified protocol fair.
>
> The metrics we reported are highly consistent with those in published models, although their reports often lack comprehensiveness. To address this, we adopted our proposed protocol, which is currently the most comprehensive available. Additionally, we attribute some discrepancies to the inherent randomness of generative models and different evaluation aspects.
>
> **Question 17:**
>
> As described in te introduction, we aim to make comparisons under the same network architecture as much as possible. Based on previous studies [2], [12], we can infer that Pocket2Mol achieves state-of-the-art performance in AR, while others, like GraphBP, perform less favorably. This led us to explore whether the lack of network representational capacity or the difference in generation strategies is the main cause of the performance disparity. Therefore, we adopted the GVP architecture used in Pocket2Mol to reach conclusions like those in Line 82-83 and Line 401-408. Otherwise, it would be meaningless to draw these conclusions regarding the strengths and weaknesses of different generation strategies.  Similarly, existing state-of-the-art one-shot SBDD models, such as MolCraft, employ a `Transformer + EGNN` network architecture. This naturally raises the question of whether their superiority over earlier models like DiffBP/DiffSBDD stems from their generative strategy or the stronger representational power of their network architecture. We believe that comparisons should be made under consistent architectures to fairly evaluate the relative strengths of different generative strategies and to highlight potential directions for future development. *Therefore, we do not view modifying the model architecture as an issue; rather, we see it as an advantage that ensures fair comparisons.*
>
> **Question 18:**
>
> All the code has been organized into a unified repository and refactored. If you are interested, please download the `supplementary.zip` file and compare it with the code repositories of previous methods. This will allow you to assess whether our engineering efforts are sufficiently comprehensive.
>
> **Question 19:**
>
> This is a protocol shared by all previous methods: generating 100 molecules for each pocket in the protein test set, resulting in 10,000 molecules across 100 pockets. In Table 2, the number 100 refers to the 100 pocket-ligand pairs in the test set. Such protocols typically do not require mathematical notation, just as one wouldn’t use  $N_{\text{class}} = 100$  to explain that CIFAR-100 has 100 classes or  $N_{\text{class}} = 1000$  to indicate that ImageNet-1k has 1000 classes. **I sincerely recommend that you review relevant prior research to familiarize yourself with the common consensus in this field (´Д｀).**
>
> **Question 21:**
>
> As stated in the paper, the generation target for each subtask corresponds to a different part of the molecule. Therefore, it is necessary to finetune the aforementioned models to better adapt them to the new downstream tasks. Additionally, for diffusion models, certain preprocessing techniques, such as center-of-mass adjustment, require re-centering the coordinate system’s origin, which necessitates retraining the model.
>
> **Question 24:**
>
> The choice of 100 is based on the aforementioned protocol, which involves generating 100 molecules for a single pocket. Additionally, for the two targets in question, the number of known chemically active molecules we collected is also around 100. Therefore, comparing the distribution of 100 generated molecules versus 100 reference molecules is reasonable.
>
> If this is still unclear, we can explain it this way: in the case study, there are two protein targets, each with approximately 100 known active molecules. For each target, we generated  100 \times \text{(number of methods)}  molecules using different methods, and then compared these generated molecules. :)

---

> ### Author Response · Authors · 2024-11-15
> **Reference**
>
> [1] Jiaqi Guan,  3D Equivariant Diffusion for Target-Aware Molecule Generation and Affinity Prediction
>
> [2] Xiangang Peng, Pocket2Mol: Efficient Molecular Sampling Based on 3D Protein Pockets
>
> [3] Wang Yidong et al. Usb: A unified semi-supervised learning benchmark. ArXiv, abs/2208.07204, 2022.
>
> [4] *Raphael Townshend et al.* ATOM3D: Tasks on Molecules in Three Dimensions, https://datasets-benchmarks-proceedings.neurips.cc/paper/2021/hash/c45147dee729311ef5b5c3003946c48f-Abstract-round1.html
>
> [5] Wangyang Wang et al, RELATION: A Deep Generative Model for Structure-Based De Novo Drug Design, https://pubs.acs.org/doi/10.1021/acs.jmedchem.2c00732
>
> [6] Mikhail Andronov et al. Exploring Chemical Reaction Space with Reaction Difference Fingerprints and Parametric t-SNE, https://pubs.acs.org/doi/10.1021/acsomega.1c04778
>
> [7] Haitao Lin et al.  Functional-Group-Based Diffusion for Pocket-SpecifiMolecule Generation and Elaboration, https://proceedings.neurips.cc/paper_files/paper/2023/file/6cdd4ce9330025967dd1ed0bed3010f5-Paper-Conference.pdf
>
> [8] Odin Zhang et al. ResGen is a pocket-aware 3D molecular generation model based on parallel multiscale modelling, https://www.nature.com/articles/s42256-023-00712-7
>
> [9] Yang Yaodong et al. Enabling target-aware molecule generation to follow multi objectives with Pareto MCTS, https://www.nature.com/articles/s42003-024-06746-w
>
> [10] Sebastian Salentin et al. PLIP: fully automated protein–ligand interaction profiler. https://pmc.ncbi.nlm.nih.gov/articles/PMC4489249/
>
> [11] Melissa F Adasme, PLIP 2021: expanding the scope of the protein–ligand interaction profiler to DNA and RNA, https://academic.oup.com/nar/article/49/W1/W530/6266421
>
> [12] Meng liu et al., Generating 3D Molecules for Target Protein Binding, https://arxiv.org/abs/2204.09410

---

> ### Author Response · Authors · 2024-11-26
> **Thanks.**
>
> Dear Reviewer G2c6:
>
> Many thanks for your raising your rating to 5. Besides, your insightful and helpful comments help us to improve our manuscript a lot. In the former discussion, we suppose we have thoroughly considered all the feedback and provided a point-to-point response to you, including the total 25 questions and 4 weakness.
>
> Since the reviewer-author discussion phase appears to have been appropriately extended, we sincerely request that you take the time to reassess our responses and the contributions of this paper. As you may have noticed, nearly all reviewers with positive opinions have provided very strong recommendations for the paper (score 8), which we believe is worth your attention as a responsible reviewer. Given that your current rating leans somewhat negative, we earnestly hope that you could re-evaluate the paper and consider providing a relatively positive recommendation. Your recognition of our work would greatly motivate us to continue our deep exploration in the SBDD field, unify frameworks, and refine engineering efforts.
>
> Best,
>
> Authors

---

### Official Review · Reviewer_g76L · 2024-11-08

**Soundness:** 3
**Presentation:** 3
**Contribution:** 3
**Rating:** 8
**Confidence:** 4

**Summary:**

This paper introduces CBGBench, addressing the gap where the absence of standardization can lead to unfair comparisons and inconclusive insights. On one hand, this paper incorporates recent state-of-the-art studies into a unified framework for fair comparisons; on the other, it extends de novo SBDD tasks to include lead optimization and other related tasks. The paper conducts extensive experiments on these tasks and provides the corresponding code.

**Strengths:**

S1. This paper has evaluated almost all the prevailing SBDD methods in generative models in AI conference.
S2. It adapt some of the models to a range of tasks essential in drug design, considered sub-tasks within the graph fill-in-the-blank tasks.
S3. It establish a training, sampling and evaluation codebase, which is comprehensive and effective affter my testing.
S4. The benchmark evaluates all the models for the two real-world target proteins, as a solid case study.

**Weaknesses:**

W1. Doubts of classification: In Line 165, it states that MolCraft using BFN as the variant of the diffusion models. However, I have two questions, Firstly, it appears that “one-shot model” in the paper refers to diffusion-based models. Given this, is it reasonable to classify BFN as a diffusion model? MolCraft generates in parameter space, while diffusion generates in data space, which I believe is a distinction. Is the authors’ classification reasonable in this regard?

W2. Doubts of evaluation metrics:  The article compares the clash ratio between protein and molecule interactions, denoted as Ratio_cca . Additionally, it defines the internal clash ratio within molecules, denoted as  Ratio_cm . Regarding the internal clash within molecules, since there are bonds connecting atoms, the defined “van der Waals radii overlap by ≥0.4Å” does not hold. For example, if this definition were applied to a benzene ring, the clash ratio would be 1.0, which fails to reflect the actual structural integrity of the molecule. Therefore, I believe this metric is unreasonable.

Minor:
Additionally, the article lacks citations for recent work related to SBDD and molecular generation, such as [1] [2]. I suggest that the authors include these references to enhance the completeness of the article.

[1] Zaixi Zhang, Mengdi Wang, Qi Lium. FlexSBDD: Structure-Based Drug Design with Flexible Protein Modeling
[2] Odin Zhang，et al. Deep Lead Optimization: Leveraging Generative AI for Structural Modification

**Questions:**

See Weakeness. The W1 and W2 are my main concern, and if a satisfactory response can be provided to the above issues, I will consider increasing the score.

---

> ### Author Response · Authors · 2024-11-14
> **Thanks! Reply to Questions and Weakness.**
>
> Thanks for your valuable comments and insightful questions, here we will reply to the two weakness with questions one by one.
>
> **Reply to W1:**
>
> Firstly, in our article, one-shot models are not limited to diffusion-based models alone. As shown in Table 1, one-shot models here refer to approaches that are in contrast to auto-regressive models. For instance, LiGAN uses a VAE model framework, which we also classify under one-shot models.
>
> Secondly, we included BFN in the diffusion model category because, in diffusion processes, the forward noising step adds noise to data using  $x_t = \alpha_t x_0 + \sigma_t \epsilon$ , which can be viewed as adding noise to the mean of a Gaussian distribution parameterized by $\mu( x_t) = \alpha_t x_0$. The reverse sampling process follows a similar logic. However, while discrete state diffusion models like [1][2] also add noise in parameter space, BFN’s update strategy for categorical variables is distinctly different, yet it can still be considered a nonlinear update process in parameter space, as opposed to the linear updating strategy with Markov transition probability in [1][2].
>
> Thus, both traditional diffusion models and BFN exhibit the following features: (1)  unification in parameter space for noise addition and sampling; and (2) progressive updates with either a linear or nonlinear strategy. Based on these similarities, we classify BFN as a diffusion-based approach.
>
> **Reply to W2:**
>
> We apologize for the lack of clarity in our explanation. We agree with your point that calculating clashes within the molecule itself is meaningless, as in structures like C=O-C, it is very likely that the distance between the two carbon atoms will be less than their van der Waals radii overlap. In fact, the term Ratio_cm here does not refer to clashes within the molecule itself; rather, it indicates the proportion of molecules with protein-molecule clashes relative to the total number of molecules. If any atom in a molecule causes a clash, we consider that molecule to have a clash. Therefore, this metric is stricter than Ratio_ca and will yield a higher ratio. We will make this explanation clearer in the next version (updated line 285) to avoid any unnecessary misunderstandings.
>
> Minor： Besides, we add the suggested reference in our updated paper. Thanks for your valuable advice.
>
>
>
> [1] [Emiel Hoogeboom](https://arxiv.org/search/stat?searchtype=author&query=Hoogeboom,+E), Argmax Flows and Multinomial Diffusion: Learning Categorical Distributions, https://arxiv.org/abs/2102.05379
>
> [2] [Andrew Campbell](https://arxiv.org/search/stat?searchtype=author&query=Campbell,+A), A Continuous Time Framework for Discrete Denoising Models, https://arxiv.org/abs/2205.14987

---

> > ### Comment · Reviewer_g76L · 2024-11-15
> > **Thanks for the rebuttal**
> >
> > Thanks for the authors's timely response. My concerns are well addressed. I would like to raise my score from 6 to 8.
> >
> > Bests,

---

> > > ### Author Response · Authors · 2024-11-15
> > > **Thanks a lot!**
> > >
> > > We sincerely appreciate your valuable feedback. Your support serves as a driving force for us to continue our efforts in this field and contribute to its advancement in the future.

---

### Official Review · Reviewer_kydR · 2024-11-09

**Soundness:** 3
**Presentation:** 4
**Contribution:** 3
**Rating:** 8
**Confidence:** 4

**Summary:**

The paper proposes CBGBench, a comprehensive benchmark for SBDD tasks, which aims to unify various generative models in a fill-in-the-blank framework for 3D complex binding graphs. It introduces a modular and extensible framework and evaluates multiple state-of-the-art methods across different metrics, including interaction, chemical properties, geometric authenticity, and substructure validity. The paper also introduces four sub-tasks: linker, fragment, side-chain, and scaffold design, to provide insights into lead optimization applications.

**Strengths:**

The paper fills a notable gap in the SBDD domain by providing a well-structured and unified benchmarking framework. The comprehensive evaluation protocol addresses the diverse nature of generative tasks.

The study uses extensive metrics to evaluate models, including metrics like Ligand Binding Efficiency (LBE) to address the size bias in generated molecules.

Application to real pharmaceutical targets like ADRB1 and DRD3 demonstrates the practical potential of the benchmark and supports the generalizability of the findings.

Overall it's a really well executed paper that focuses on an particular case of generative models, which is drug design, but deserved an acceptance to the main conference due to the relevance of the task.

**Weaknesses:**

Further testing on more diverse real world systems should be done. Exploring how this models behave in systems like KRAS12 for instance where the main goal is growing into subpockets, would enrich this study.

**Questions:**

Include more systems that showcase corner cases for a benchmark that are common in real scenarios. i.e KRAS12, BRD4

The comments were satisfactorily addressed.

---

> ### Author Response · Authors · 2024-11-14
> **Thanks! Reply and extra test on KRAS12**
>
> We sincerely appreciate your recognition of our work. In response to the issues you raised, we provide the following replies:
>
> Following [1], we focused on the two targets ADRB1 and DRD3, both of which play significant regulatory roles in biological systems, making case studies on these targets meaningful.
>
> Additionally, based on your valuable feedback, we recently included experiments with the KRAS12 target (PDB code: 7O83) and used the bound structure’s protein configuration as the pocket (see Fig. 10 in the Appendix E3 in the latest article version). Using the methods incorporated in our case study, we generated 100 molecules for the target, then assessed their molecular fingerprint distribution as well as Vina Energy & LBE. The results have been added to Appendix E3 in Figures 11(a) and (b). Furthermore, we provided a summary table of Mean & Median values related to binding affinity of the generated molecules on KRAS12 for brief reading.
>
> | Method | E_vina Mean | E_vina Median | LBE Mean | LBE Median |
> | --- | --- | --- | --- | --- |
> | Pocket2Mol | -7.44 | -7.43 | 0.373 | 0.342 |
> | GraphBP | -3.91 | -3.72 | 0.268 | 0.255 |
> | TargetDiff | -9.29 | -9.29 | 0.334 | 0.337 |
> | DiffSBDD | -7.67 | -7.50 | 0.297 | 0.284 |
> | DiffBP | -7.46 | -7.57 | 0.294 | 0.297 |
> | FLAG | -5.35 | -5.05 | 0.267 | 0.271 |
> | D3FG | -8.73 | -8.60 | 0.341 | 0.326 |
> | MolCraft | -9.94 | -9.67 | 0.344 | 0.347 |
>
> [1] Haotian Zhang, Delete: Deep Lead Optimization Enveloped in Protein Pocket, https://arxiv.org/pdf/2308.02172

---

> ### Author Response · Authors · 2024-11-23
> **Response**
>
> Dear Reviwer kydR:
>
> We sincerely and deeply appreciate your efforts and valuable advice. It is the dedication and active engagement of reviewers like you, who continuously engage in discussions with authors, that drive the development of the ML community and the AI4Science community.
>
> As you can see, we have added extensive parts for real-world target analysis, including KRAS12 as you suggested. Since the reply has been made from about one week before, and there will be about three days left until the discussion deadline, we would like to know if we have addressed all your concerns at this stage. If you have any further related concerns, please feel free to share them with us, and we will be more than happy to address them. If you are satisfied with our responses, we kindly ask if you would consider reevaluating the manuscript and providing a stronger recommendation. If you recognize the value of our work and the efforts we have made during the rebuttal phase, we kindly ask you to consider raising the score to ‘Accept’, as two of the reviewers has raised their rating. We would be immensely grateful, as this would acknowledge the significant effort we have invested. In the future, we will expand our framework and codebase with the newest advancements, making ongoing contributions to the AI for drug design community.
>
> Your feedback, recognition of our work, and recommendation are extremely important to us. :)

---

> ### Author Response · Authors · 2024-11-25
> **Many Thanks.**
>
> Dear Reviewer kydR:
>
> Thank you for your insightful and helpful comments once again. We suppose we have thoroughly considered all the feedback and provided a point-to-point response to you, espicially add the target of KRAS12 as a case study.
>
> The deadline for author-reviewer discussions is approaching. If you still need any clarification or have any other questions, please do not hesitate to let us know.  We hope that our response addresses your concerns to your satisfaction and we would like to kindly request a reconsideration of a higher rating if it is possible.
>
> Best regards,
>
> Authors.

---

> > ### Comment · Reviewer_kydR · 2024-11-25
> >
> > Dear authors,
> >
> > Thank you for your extra work. I reassessed the score to a 8.

---

> > > ### Author Response · Authors · 2024-11-26
> > > **Thanks!**
> > >
> > > Thanks for your recognition of our works! Your insightful and helpful suggestions help us to improve our works a lot. Your support serves as a driving force for us to continue our efforts in this field and contribute to its advancement in the future.

---

### Official Review · Reviewer_Fmfm · 2024-11-11

**Soundness:** 3
**Presentation:** 3
**Contribution:** 3
**Rating:** 8
**Confidence:** 4

**Summary:**

This paper proposes a benchmark for SBDD, including a unified framework of generative graph completion for multiple tasks in the field and a comprehensive evaluation protocol.

**Strengths:**

1. The unified code base is a nice contribution to the community and beneficial for future research.
2. The evaluation protocol is comprehensive with a reasonable benchmark setting.
3. The benchmarked methods are representative and state-of-the-art.

**Weaknesses:**

1. It would be better if the author could discuss the recent trend of training a unified model for small molecules and macromolecules such as proteins and nucleic acids, and its implications on the field of SBDD.

**Questions:**

1. Why do you choose different GNN architectures for auto-regressive and diffusion-based models?
2. MolCraft seems to be missing from the t-SNE visualization in Figure 8. Also, the distributions of Vina Dock Energy and LBE are not provided. Could you please provide these results?
3. [credit to Associate PC] How might your benchmark and evaluation protocol need to be adapted to assess unified models that can generate both small molecules and macromolecules like proteins?

---

> ### Author Response · Authors · 2024-11-14
> **Thanks! Reply to the questions.**
>
> Thanks for your recognition of the value of our work. Here is our reply to your questions.
>
> **Reply to Q1**:
>
> We here use GVP as the backbone architectures for Auto-regressive models and EGNN as the architecture for One-shot models, for the two reasons:
>
> 1. GVP projects the vector (positions) into a higher dimensional space, and update the latent coordinates, while EGNN directly update the 3D Euclidean coordinates. From this perspective, GVP requires more memory and computational comlexity compared to EGNN. This is because GVP projects both coordinates and attributes into a high-dimensional space. This makes GVP more feasible in an auto-regression (AR) model, where the AR model only needs to model the coordinates of one atom (the next atom) at a time. However, in one-shot methods, such as diffusion, it must model the joint distribution of the coordinates and types of  $N_{\text{atom}}$  atoms simultaneously, leading to potentially unaffordable computational costs. Therefore, we chose the GNN architecture  with the strongest possible representational capacity within feasible conditions, adapting it to different generation methods to minimize the negative impact of insufficient network expressiveness on performance and to explore various generation strategies and methods.
> 2. Moreover, as described in the introduction, we aim to make comparisons under the same network architecture as much as possible. Based on previous studies [1], [2], we can infer that Pocket2Mol achieves state-of-the-art performance in AR, while others, like GraphBP, perform less favorably. This led us to explore whether the lack of network representational capacity or the difference in generation strategies is the main cause of the performance disparity. Therefore, we adopted the GVP architecture used in Pocket2Mol to reach conclusions like those in Line 82-83 and Line 401-408. Otherwise, it would be meaningless to draw these conclusions regarding the strengths and weaknesses of different generation strategies.
>
> **Reply to Q2**:
>
> We are sorry about the newest revision on the experimental results in the Appendix is not updated. We give the result in Figure 8 and 9 in the Appendix of the latest updated version. The t-SNE visualization distribution changed since the MolCraft molecule is added and making the training data for t-SNE different. Thanks a lot for your mentioning our mistake.
>
> **Reply to Q3&Weakness**:
>
> Since this paper primarily focuses on the SBDD task, we have not discussed broader biomolecular generation tasks. Here, we can provide an overview of this direction. A common consensus is that **de novo** biomolecule design requires models to generate both the types and structures of basic units in biomolecules.
>
> Therefore, from a structural biology perspective, these basic units are amino acids in proteins and peptides, bases in DNA, and atoms in small molecules. Although it is possible to generate these biomolecules atom-by-atom from the bottom up, the inherent “biological language” of proteins (e.g., AFCUDNE…) and DNA (e.g., ATCG) complicates ensuring that atomic-level generation can successfully assemble corresponding amino acids or bases. Consequently, bottom-up generation is challenging for other macro biomolecules. In contrast, a top-down generation strategy that treats functional groups as fundamental units in proteins and amino acids can inspire molecular design and SBDD tasks. Examples of such approaches include [3] and [4].
>
> Besides, from a model design perspective, as the fundamental generation units for proteins and other biomolecules are functional groups, determining the atomic coordinates within each functional group becomes an essential issue in transitioning from coarse-grained to fine-grained generation. Current models often treat the protein backbone as a rigid body and side chains as flexible elements, generating the C-alpha positions and orientations of amino acids in the backbone. This approach has inspired the motivations in SBDD tasks like [3] and the generation strategies in lead optimization like [5].
>
> Overall, while no model currently unifies **de novo** design for both marco-biomolecules and small molecules, the fields are increasingly converging. For instance, AlphaFold3 focuses on protein&DNA&RNA-ligand bound structure prediction (though not yet **de novo** design), and research on unified approaches is expected to make steady progress in the coming years.
>
> [1] Meng liu et al., Generating 3D Molecules for Target Protein Binding
>
> [2] Xiangang Peng et al., Pocket2Mol: Efficient Molecular Sampling Based on 3D Protein Pockets
>
> [3] Haitao Lin et al, Functional-Group-Based Diffusion for Pocket-Specific Molecule Generation and Elaboration
>
> [4] Jiaqi Guan et al, DECOMPDIFF: Diffusion Models with Decomposed Priors
> for Structure-Based Drug Design
>
> [5] Jiaqi Guan et al., LinkerNet: fragment poses and linker co-design with 3D equivariant diffusion

---

> > ### Author Response · Authors · 2024-11-26
> > **Thanks for review.**
> >
> > Dear Reviewer Fmfm:
> >
> > Thank you for your insightful and helpful comments once again. We suppose we have thoroughly considered all the feedback and provided a point-to-point response to you. If you still need any clarification or have any other questions, please do not hesitate to let us know.
> >
> > Best regards,
> >
> > Authors.

---

> > > ### Comment · Reviewer_Fmfm · 2024-11-26
> > > **Response to the authors**
> > >
> > > Thank you for your time and effort spent in the revision of the paper. I am happy to maintain my score of 8.

---

### Meta-Review · Area_Chair_WNCa · 2024-12-19

**Metareview:**

The paper makes a solid contribution by introducing a novel benchmark for models applied to structure-based drug discovery.

Reviewers appreciated the scope of the evaluated methods, the design of the subtasks, and the shared codebase. All reviewers voted to accept the paper.

I agree with the reviewers and I am happy to recommend the work for the ICLR program.

**Additional Comments On Reviewer Discussion:**

During the rebuttal, reviewers highlighted concerns about the clarity of task definitions, justification for chosen metrics, and the novelty of contributions in relation to existing benchmarks. The authors addressed these by refining the task descriptions, adding explanations for evaluation protocols like Ligand Binding Efficiency (LBE), and incorporating new experiments, including real-world applications such as KRAS12 case studies. All in all, the paper was substantially improved during the discussion phase.

---

### Decision · Program_Chairs · 2025-01-22

Accept (Spotlight)